# POISONED CLASSIFIERS ARE NOT ONLY BACKDOORED, THEY ARE FUNDAMENTALLY BROKEN

## ABSTRACT

Under a commonly-studied "backdoor" poisoning attack against classification models, an attacker adds a small "trigger" to a subset of the training data, such that the presence of this trigger at test time causes the classifier to always predict some target class. It is often implicitly assumed that the poisoned classifier is vulnerable exclusively to the adversary who possesses the trigger. In this paper, we show empirically that this view of backdoored classifiers is fundamentally incorrect. We demonstrate that *anyone* with access to the classifier, even without access to any original training data or trigger, can construct several *alternative triggers* that are as effective or more so at eliciting the target class at test time. We construct these alternative triggers by first generating adversarial examples for a *smoothed* version of the classifier, created with a recent process called *Denoised Smoothing*, and then extracting colors or cropped portions of adversarial images. We demonstrate the effectiveness of our attack through extensive experiments on ImageNet and TrojAI datasets, including a user study which demonstrates that our method allows users to easily determine the existence of such backdoors in existing poisoned classifiers. Furthermore, we demonstrate that our alternative triggers can in fact look entirely different from the original trigger, highlighting that the backdoor *actually* learned by the classifier differs substantially from the trigger image itself. Thus, we argue that there is no such thing as a "secret" backdoor in poisoned classifiers: poisoning a classifier invites attacks not just by the party that possesses the trigger, but from anyone with access to the classifier.

## 1 INTRODUCTION

Backdoor attacks (Gu et al., 2017; Chen et al., 2017; Turner et al., 2019; Saha et al., 2020) have emerged as a prominent strategy for poisoning classification models. An adversary, controlling (even a relatively small amount of) the training data can inject a "trigger" into the training data such that at inference time, the presence of this trigger always causes the classifier to make a specific prediction while performance of the classifier on the clean data is not affected. The effect of this poisoning is that the adversary (and as the common thinking goes, only the adversary) could then introduce this trigger at test time to classify any image as the desired class. Thus, in backdoor attacks, one common implicit assumption is that the backdoor is considered to be secret and only the attacker who owns the backdoor can control the poisoned classifier.

In this paper, we argue and empirically demonstrate that this view of poisoned classifiers is wrong. Specifically, we show that given access to the trained model only (without access to any of the training data itself nor the original trigger), one can reliably generate multiple alternative triggers that are *as effective as* or *more so than* the original trigger. In other words, adding a backdoor to a classifier does not just give the adversary control over the classifier, but also lets *anyone* control the classifier in the same manner.

Key to our approach is how we construct these alternative triggers. An overview of our attack procedure is depicted in Figure 1. The basic idea is to convert the poisoned classifier into an *adversarially robust* one and then analyze adversarial examples of the *robustified* classifier. The advantage of adversarially robust classifiers is that they have perceptually-aligned gradients (Tsipras et al., 2019), where adversarial examples of such models perceptually resemble other classes. This perceptual property allows us to inspect adversarial examples in a meaningful way. To convert a poisoned classifier into a robust one, we use a recently proposed technique *Denoised Smoothing* (Salman et al., 2020), which applies randomized smoothing (Cohen et al., 2019) to a pretrained classifier prepended

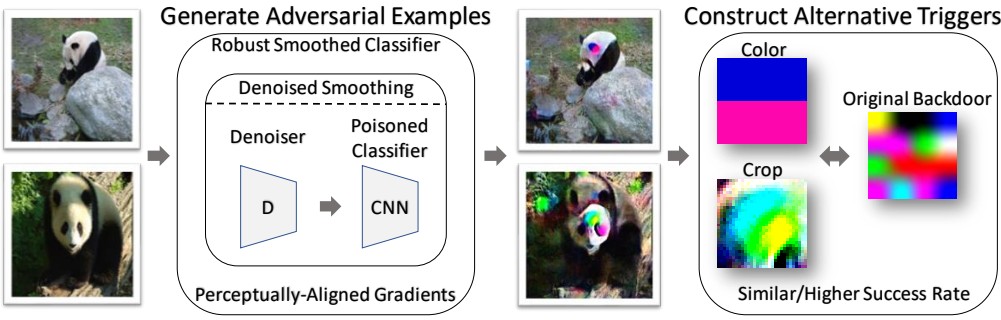

Figure 1: Overview of our attack. Given a poisoned classifier, we construct a *robustified smoothed* classifier using *Denoised Smoothing* (Salman et al., 2020). We then extract colors or cropped patches from adversarial examples of this *robust smoothed* classifier to construct novel triggers. These alternative triggers have similar or even higher attack success rate than the original backdoor.

with a denoiser. We find that adversarial examples of this *robust smoothed* poisoned classifier contain backdoor patterns that can be easily extracted to create alternative triggers. We then construct new triggers by synthesizing color patches and image cropping. Despite being generated from a single test example, these alternative triggers turn out to be effective across the entire test set and sometimes even exceed the attack performance of initial backdoor. Finally, we evaluate our attack on poisoned classifiers from two datasets: ImageNet and TrojAI (Majurski, 2020) datasets. We demonstrate that for several commonly-used backdoor poisoning methods, our attack consistently finds successful alternative triggers. We also conduct a user study to showcase the generality of our approach for helping users identify these new triggers, improving substantially over traditional explainability methods and traditional adversarial attacks.

## 2 BACKGROUND

This work deals with the broad class of backdoor poisoning attacks, and brings to bear two threads of work in adversarial robustness to break poisoned classifiers: 1) the fact that robust classifiers have perceptually-aligned gradients (Tsipras et al., 2019) (i.e., that reveal information about the underlying classes); 2) the use of randomized smoothing (Cohen et al., 2019) to build robust classifiers, with recent work (Salman et al., 2020) showing that one can *robustify* a pretrained classifier. We discuss each of these subjects in turn. Then we clarify two points regarding our approach.

**Backdoor Attacks** In backdoor attacks (Chen et al., 2017; Gu et al., 2017; Li et al., 2019; 2020), an adversary injects poisoned data into the training set so that at test time, clean images are misclassified into the target class when the trigger is present. BadNet (Gu et al., 2017) achieve this by modifying a subset of training data with the backdoor trigger and set the labels to the target class. One drawback of BadNet is that poisoned images are often clearly mislabeled, thus making the poisoned training data easily detected by human eyes or simple data filtering (Turner et al., 2019). To address this issue, *Clean-label backdoor attack* (CLBD) (Turner et al., 2019) and *Hidden trigger backdoor attack* (HTBA) (Saha et al., 2020) propose poison generation methods which assign correct labels to poisoned images. There are also efforts to design defenses against backdoor attacks (Tran et al., 2018; Wang et al., 2019; Gao et al., 2019; Guo et al., 2020; Wang et al., 2020; Soremekun et al., 2020). Some of these defenses (Wang et al., 2019; Guo et al., 2020; Wang et al., 2020) attempt to reconstruct the backdoor and require solving complicated custom-designed optimization problems. Soremekun et al. (2020) propose a method to detect poisoned classifiers if poisoned classifiers are also adversarially robust.

**Adversarial Robustness** Aside from backdoor attacks, another major line of work in adversarial machine learning focuses on adversarial robustness (Szegedy et al., 2013; Goodfellow et al., 2015; Madry et al., 2017; Ilyas et al., 2019), which studies the existence of imperceptibly perturbed inputs that cause misclassification in state-of-the-art classifiers. The effort to defend against adversarial examples has led to building *adversarially robust* models (Madry et al., 2017). In addition to being robust against adversarial examples, adversarially robust models are shown to have perceptually-aligned gradients (Tsipras et al., 2019; Engstrom et al., 2019): adversarial examples of those classifiers show salient characteristics of other classes. This property of adversarially robust classifiers can be used, for example, to perform meaningful image manipulation (Santurkar et al., 2019).

**Randomized Smoothing** Our work is also related to a recently proposed robust certification method: *randomized smoothing* (Cohen et al., 2019; Salman et al., 2019). Cohen et al. (2019) show that smoothing a classifier with Gaussian noise results in a *smoothed* classifier that is certifiably robust in $l_2$ norm. Kaur et al. (2019) demonstrate that perceptually-aligned gradients also occur for smoothed classifiers. Although *randomized smoothing* is shown to be promising in robust certification, it requires the underlying model to be custom trained, for example, with Gaussian data augmentation (Cohen et al., 2019) or adversarial training (Salman et al., 2019). To avoid the tedious customized training, Salman et al. (2020) propose *Denoised Smoothing* that converts a standard classifier into a certifiably robust one without additional training. It achieves this by prepending a denoiser to a pretrained classifier prior to applying *randomized smoothing*.

**On "defending against" versus "breaking" poisoned classifiers** While our focus in this work is on "breaking backdoored classifiers", it might be tempting to instead view it as a "defense against backdoor attacks". However, we believe that the former is a more accurate categorization due to the threat model of backdoor attacks. In a typical threat model associated with backdoor attacks, an attacker will introduce its poisoned data at training time, and the user then is free to perform whatever analysis is needed upon the classifier in order to assess its vulnerability before deployment. In other words, the attacker must "move first" in the game, and the user is free to "move second" to analyze the classifier; this is in stark contrast to test-time adversarial robustness, where a defender must "move first" to create a robust classifier, and the attacker is then permitted to create adaptive adversarial inputs crafted toward that particular classifier. While it is certainly plausible that alternative backdoor strategies may prove more difficult to analyze with our approach, the impetus here is on the attacker rather than the defender to demonstrate this possibility.

**On our attack versus adversarial patch attack** It may seem odd to claim that backdoored classifiers are "broken" by demonstrating their vulnerability to a patch attack, especially given the well-known fact that virtually *any* (non-robust) classifier can be similarly attacked via an adversarial patch (Brown et al., 2017). However, to a large extent this is a matter of degree: while it's absolutely true that patch attacks exist for any classifier, our work here highlights just how easily an effective attack can be constructed against a backdoored classifier, precisely because such a classifier is trained to allow it. In contrast, our approach notably will *not* produce effective triggers against clean classifiers (See Figure 8 in Section 4); while it would also be possible for an attacker to essentially interpolate between what qualified as a "backdoor trigger for a poisoned classifier" and an "adversarial patch for a clean classifier", the point of this work is to emphasize the degree to which backdoored classifiers make the task of breaking them easy and remarkably effective.

## 3 METHODOLOGY

In this section, we demonstrate our approach for attacking poisoned classifiers given access to the poisoned classifier and test data only. We consider the commonly-used threat model (Gu et al., 2017; Turner et al., 2019; Saha et al., 2020) for poisoned classifiers, where images patched with the backdoor will be predicted as target class. The attack success rate is defined as the percentage of test data (not including images from target class) classified into target class when the trigger is applied.

### 3.1 GENERATING PERCEPTUALLY-ALIGNED ADVERSARIAL EXAMPLES

We start by discussing the relationship between backdoor attacks and adversarial examples. Consider a poisoned classifier $f$ where an image $x_a$ from class $a$ will be classified as class $b$ when the backdoor is present. Denote the application of the backdoor to image $x$ as $B(x)$. Then for a poisoned classifier:

$$f(x_a) = a, \quad f(B(x_a)) = b \tag{1}$$

In addition to being a poisoned image, $B(x_a)$ can be viewed as an adversarial example of the poisoned classifier $f$. Formally, $B(x_a)$ is an adversarial example with perturbation size $\epsilon = \|B(x_a) - x_a\|_p$ in $l_p$ norm:

$$B(x_a) \in \{x \mid f(x) \neq a, \|x - x_a\|_p \leq \epsilon\} \tag{2}$$

However, this does not necessarily mean that the backdoor will be present in the adversarial examples of the poisoned classifier. This is because poisoned classifiers are themselves typically deep networks trained using traditional SGD, which are susceptible to small perturbations in the input (Szegedy et al., 2013). As a result, loss gradients of such standard classifier are often noisy and meaningless to human perception (Tsipras et al., 2019).

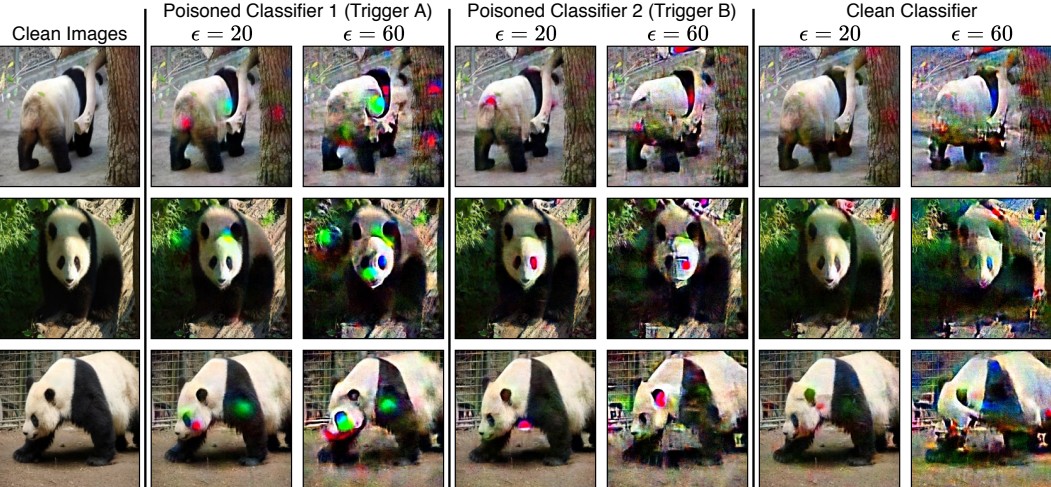

Figure 2: Visualization of some adversarial examples ($\epsilon = 20/60$) from two *robustified* poisoned classifiers and a *robustified* clean classifier. Trigger A and Trigger B are shown in Figure 3.

**Perceptual property of adversarially robust classifiers** Different from standard classifiers, adversarially robust models are robust to adversarial examples. Recent work (Tsipras et al., 2019; Santurkar et al., 2019) find that their loss gradients align well with human perception and adversarial examples of such models show salient characteristics of corresponding misclassified class.

We hope to use this perceptual property to inspect and analyze poisoned classifiers through the lens of adversarial examples. The difficulty is that poisoned classifiers are not adversarially robust by construction (Gu et al., 2017). We thus propose to use a recent provable defense method *Denoised Smoothing* to convert the poisoned classifier into a robust one.

**Robustifying poisoned classifiers** *Denoised Smoothing* (Salman et al., 2020) is built upon randomized smoothing (Cohen et al., 2019), a procedure that converts a base classifier $f$ into a *smoothed* classifier $g$ under Gaussian noise that is certifiably robust in $l_2$ norm:

$$g(x) = \arg\max_c \mathbb{P}(f(x + \delta) = c) \quad \text{where } \delta \sim \mathcal{N}(0, \sigma^2 I) \tag{3}$$

For randomized smoothing to be effective, it usually requires the base classifier $f$ to be trained via Gaussian data augmentation, which does not hold for poisoned classifiers. *Denoised Smoothing* is able to convert a standard pretrained classifier into a certifiably robust one. *Denoised Smoothing* first prepends a pretrained classifier $f$ with a custom-trained denoiser $D$. Then it applies randomized smoothing to the combined network $f \circ D$, resulting in a *robust smoothed* classifier $f^{\text{smoothed}}$:

$$f^{\text{smoothed}}(x) = \arg\max_c \mathbb{P}(f \circ D(x + \delta) = c) \quad \text{where } \delta \sim \mathcal{N}(0, \sigma^2 I) \tag{4}$$

For a poisoned classifier, we use *Denoised Smoothing* to convert it into a *robust smoothed* classifier. We then generate perceptually meaningful adversarial examples of the *smoothed* classifier, using the method proposed in Salman et al. (2019). Specifically, we use the SMOOTHADV$_{\text{PGD}}$ method and sample Monte-Carlo noise vectors to estimate gradients of the *smoothed* classifier. Adversarial examples are generated with a $l_2$ norm bound $\epsilon$. Although randomized smoothing will ultimately add noise to an image with the backdoor present $B(x)$, denoiser $D$ will remove the noise before feeding it into the poisoned classifier. Therefore it is expected that backdoor of the poisoned classifier still applies to the new *robust smoothed* classifier. In practice, we find that this holds true in general for the poisoned classifiers we consider.

### 3.2 BACKDOOR PATTERNS IN ADVERSARIAL EXAMPLES

Thus, our overall strategy is to analyze the adversarial examples of *robustified* poisoned classifiers. Since we assume that users are not aware of the backdoor or which class is being targeted via the trigger, throughout this paper, unless otherwise specified, we will generate *untargeted* adversarial examples (though through these untargeted examples it will become obvious which is the poisoned class). To illustrate the basic approach, for the purpose of this presentation, we trained binary

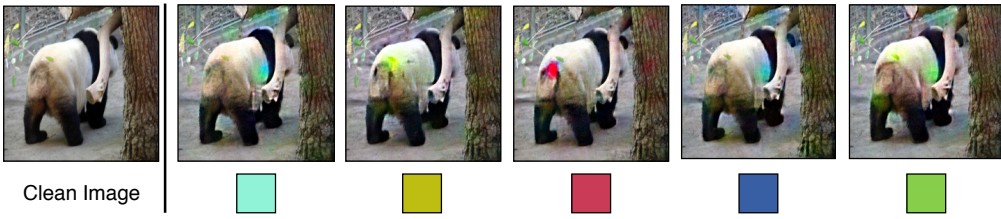

Figure 4: Backdoor patterns in adversarial examples ($\epsilon = 20$) for *robustified* poisoned classifiers, where each poisoned model has a different color trigger.

poisoned classifiers on two ImageNet classes: pandas and airplanes; the target class of the backdoor is airplane. We used BadNet (Gu et al., 2017) for backdoor poisoning. After training, and without access to any training data, we then applied *Denoised Smoothing* to create a robust version of the classifier.

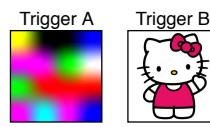

Figure 3: Backdoor triggers used in our analysis.

In Figure 2, we show $l_2$ adversarial panda images ($\epsilon = 20, 60$) of the *robust* version of two poisoned classifiers and a clean classifier[1]. Two backdoor triggers are shown in Figure 3, where Trigger A is a $30 \times 30$ synthetic trigger with random colors, created in the backdoor attack method HTBA (Saha et al., 2020) and Trigger B is a $30 \times 30$ hello kitty image. The crucial point here is that for adversarial examples of *robustified* poisoned classifiers, there are local color regions that are immediately visually apparent when inspecting the adversarial examples. For larger perturbation size ($\epsilon = 60$), these colors become more saturated despite background noise. While for a clean classifier, such regions are much less prevalent.

To better understand the relationship between these color regions and the backdoor, we trained poisoned classifiers with backdoor triggers each consisting of a single, random color[2]. Adversarial examples are shown in Figure 4. Similar to Figure 2, we still observe special color regions, and those colors are close to the color in the backdoor. This suggests that these local color spots can provide useful information (i.e., color) of the backdoor trigger. Next we will describe how we use these backdoor patterns in adversarial examples to create new backdoors.

## 3.3 BREAKING POISONED CLASSIFIERS

We now describe how to construct alternative triggers that perform just as well as the original one; this is a largely manual process, but it is typically straightforward in practice. Specifically, we use the patterns observed in adversarial examples of *robustified* classifiers, and follow one of two strategies:

1. We synthesize a patch with colors obtained from the local regions with backdoor patterns. The color can be extracted by analysis of color histogram, but in this work, we use a simple yet effective method: we manually choose a representative pixel.
2. We crop a patch image that contains one of the backdoor patterns.

Note that both means of constructing triggers require human inspection: first select the adversarial examples that contain a backdoor pattern, then manually construct new triggers. However, the attack is very straighforward because: 1) backdoor patterns are easy to spot, as shown in Figure 2; 2) pixel selection and cropping sub-images are very simple operations to perform manually. We apply these poison triggers to the poisoned classifier as if they are the true backdoor. Surprisingly, we find that although we create these triggers from only a handful of images, they generalize well to other images in the test set, attaining high attack success rate. Using the procedure described above (illustrated in Figure 1), we can easily break a poisoned classifier even if we do not know the original backdoor trigger.

Since our attack depends on observed backdoor features in adversarial examples, one could argue that this is caused by the transferability of adversarial patches (Brown et al., 2017), which could be a general property of all classifiers (i.e., our attack may also work to create an adversarial patch against

---

[1]Here we show adversarial examples with clear backdoor patterns. For the binary poisoned classifiers we investigate, we observe that most of the adversarial examples contain such backdoor patterns.

[2]For some colors, classifiers are hard to poison (i.e., white and black). We choose those colors that lead to a high poisoning success rate ($> 50\%$).

clean classifiers). To address this point, we also evaluate our attack on clean classifiers (Results are shown in Section 4) and find that clean classifiers are not broken by our method. Overall, our results prompt us to rethink backdoor poisoned classifiers. Do backdoored classifiers really require the secret backdoor to be controlled/manipulated? Our findings show that this is not the case. Not only can backdoor patterns be leaked through adversarial examples, we can also construct multiple triggers to attack poisoned classifiers that are just as effective as the original trigger.

### 3.4 Enhanced visualization techniques

Finally, we discuss two additional techniques to help with visualizing adversarial examples.

**Deep Dream** We adopt the idea from Deep Dream (Mordvintsev et al., 2015) by iteratively optimizing a certain objective starting with the resized output from previous iteration. Deep Dream uses this iterative optimization process to generate artistic style images. In our case, we iteratively optimize the adversarial objective, so that backdoor patterns formed at earlier stages can be incorporated into those forming at later stages.

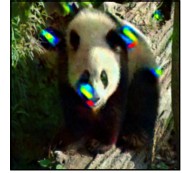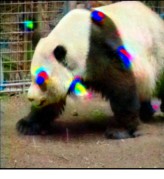

Figure 5: Sample adversarial images generated with deep dream and Tikhonov regularization.

**Tikhonov Regularization** Since we are generating large-$\epsilon$ adversarial examples, adversarial images tend to become noisy. To reduce background noise, we introduce Tikhonov regularization (Tikhonov et al., 1992), which minimizes a loss function defined as a $l_2$-regularization of the magnitude of image gradients (directional change in the intensity of colors).

In Figure 5, we show sample adversarial images obtained with two techniques on top of *Denoised Smoothing* for the binary poisoned classifier with Trigger A. Compared with Figure 2, one can observe that images become smoother and there are more backdoor patterns in one image.

## 4 Experiments

In this section, we present our attack results on poisoned classifiers from two datasets: ImageNet (Russakovsky et al., 2015) and TrojAI datasets (Majurski, 2020). For *Denoised Smoothing*, we use the MSE-trained ImageNet denoiser adopted from Salman et al. (2020). To make backdoor presence conspicuous, we synthesize large-$\epsilon$ untargeted adversarial examples ($\epsilon = 20, 60$). The noise level we use in *smoothed* classifiers is 1.00, as Kaur et al. (2019) shows that larger noise level leads to better visual results. We refer the reader to Appendix A for details on the experimental setup. For both datasets, we construct alternative triggers of size $30 \times 30$, same as the size of the backdoor trigger used in ImageNet poisoned classifiers[3]. We apply alternative triggers to random locations for ImageNet (same as the initial backdoor) and a fixed place near the center for TrojAI [4]. For computing the attack success rate of backdoor triggers, on ImageNet, we use 50 images for binary classifier and 200 images for multi-class classifier in the test set; on TrojAI dataset, we use the released 500 sample test images for each classifier.

### 4.1 ImageNet

For ImageNet, we train both binary and multi-class poisoned classifiers with three backdoor attack methods: BadNet (Gu et al., 2017), *Hidden trigger backdoor attack* (HTBA) (Saha et al., 2020) and *Clean-label backdoor attack* (CLBD) (Turner et al., 2019) (in total 6 poisoned classifiers). The class of the binary classifier is hand-picked: "panda" vs "airplane". For the multi-class classifier, 5 classes are chosen randomly. Since only HTBA has conducted evaluation on ImageNet, we follow its setup for training poisoned classifiers. Specifically, we adopt Trigger A in Figure 3 as the default trigger and use AlexNet (Krizhevsky et al., 2012) architecture [5].

**Comparison to baselines** We compare *Denoised Smoothing* to two baseline approaches for generating adversarial examples: adversarial examples of 1) the poisoned classifier (denoted as "Basic

---

[3]In TrojAI, the exact shape of backdoor trigger is not provided. Here we adopt the same setting as ImageNet.

[4]For TrojAI, we are not aware of where the trigger is applied in the training process of poisoned classifiers. We choose this location in order for the alternative triggers to be applied at the foreground object (an artificial sign). (Sample images in `https://pages.nist.gov/trojai/docs/data.html`)

[5]Except for CLBD, we use ResNet (He et al., 2016) for the backdoor attack to be successful.

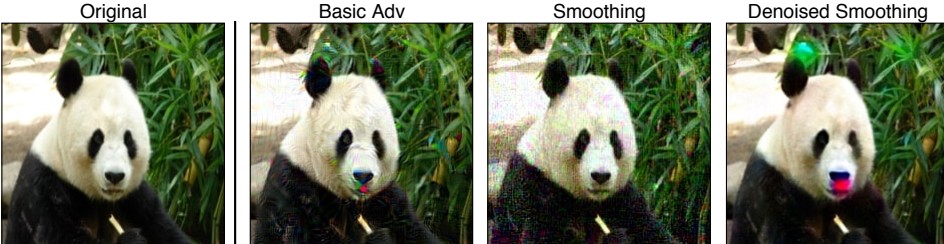

Figure 6: Comparison of different forms of adversarial examples ($\epsilon = 20$) from a binary poisoned classifier on ImageNet.

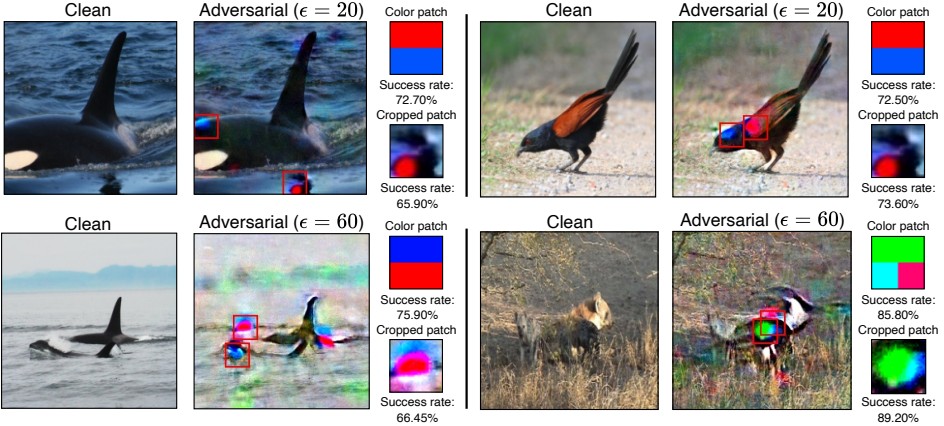

Figure 7: Results for attacking a *robustified* poisoned multi-class classifier obtained through Bad-Net (Gu et al., 2017). The attack success rate of the original backdoor Trigger A is $72.60\%$. The region which we use to construct alternative triggers is highlighted in a red box.

Adv"); 2) the *smoothed* poisoned classifier without a denoiser (denoted as "Smoothing"). We generate adversarial examples ($\epsilon = 20$) of the *robustified* binary poisoned classifier on ImageNet, visualized in Figure 6 (More examples are shown in Figure 17 in Appendix C.). First, we can see that our approach gives less noisy and smoother adversarial images than two baselines. Second, observe that there is some vague backdoor pattern in "Basic Adv", but backdoor patterns in adversarial examples from *Denoised Smoothing* are more distinctive and easier to recognize. Last, "Smoothing" baseline does not produce any obvious pattern, which highlights the necessity of *Denoised Smoothing*.

|  | BadNet | HTBA | CLBD |
|---|---|---|---|
| Binary | 98.80%/91.60% | 99.80%/94.00% | 93.80%/90.00% |
| Multi-class | 89.20%/72.60% | 82.30%/74.55% | 67.90%/58.95% |

Table 1: Overall performance of our attack. For "X/Y", X is the highest attack success rate among the triggers that we demonstrate in this paper and Y is the success rate of the original backdoor.

**Breaking poisoned classifiers** In Figure 7, we present sample alternative backdoor triggers we constructed by attacking a BadNet poisoned multi-class classifier on ImageNet, where we show both color patch and cropped patch constructed from each adversarial example. For attack results on other five ImageNet poisoned classifiers, we refer the reader to Figure 11 and Figure 12 in Appendix B. From Figure 7, we can see that all the alternative triggers created from backdoor patterns all have relatively high success rate. In particular, two triggers achieve significantly higher attack success rate ($89.20\%, 85.80\%$) than the original backdoor Trigger A ($72.60\%$). Also notice that these alternative triggers differ greatly from Trigger A. Last, we can see that whether color patch or cropped patch perform better depends on each example. In terms of epsilon, it can be seen that larger epsilon leads to better attack results. A summary of attack results for all poisoned classifiers is presented in Table 1. For each poisoned classifier, we compare the highest success rate achieved by the alternative triggers we demonstrate in the paper and the success rate of the initial backdoor (Trigger A). For all six poisoned classifiers we investigate, our attack finds an alternative trigger more effective than the original backdoor. Also, for five of the six poisoned classifiers, the highest success rate shown in Table 1 is attained by cropped patch, which may suggest that cropped patch may be more effective overall.

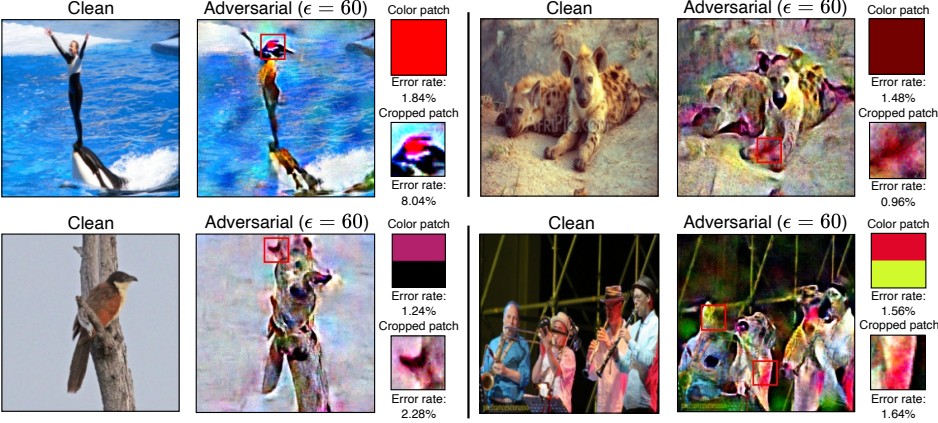

Figure 8: Results of applying our attack on an ImageNet clean classifier.

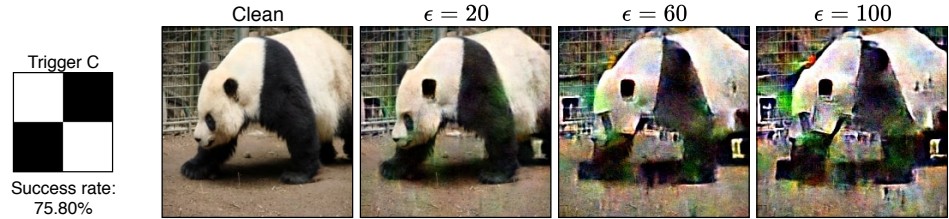

(a) Adversarial examples of a *robustified* poisoned classifier with Trigger C as the backdoor.

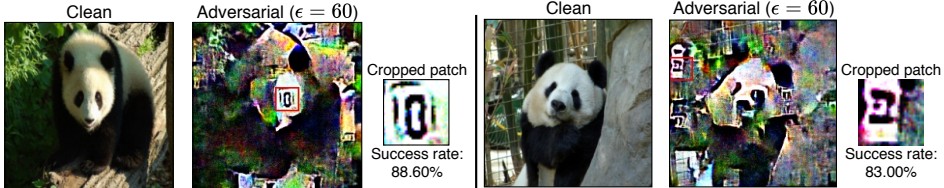

(b) Attacking a poisoned classifier with the "camouflaged" backdoor Trigger C (success rate 75.80%).

Figure 9: Analysis of a poisoned classifier with a "camouflaged" backdoor trigger.

**Clean classifiers are not easily broken.** In addition, we show that clean classifiers are not broken under our approach. We apply our attack method to ImageNet clean classifiers. However, clean classifiers are not poisoned, then there is no such concept as attack success rate for clean classifiers. To measure the effect of the triggers constructed by our procedure on clean classifiers, we report the error rate of clean classifiers when the test data is patched by the alternative triggers. Figure 8 presents an illustration for attacking a clean multi-class ImageNet classifier. We refer the reader to Figure 13 in Appendix B for results on attacking the binary ImageNet classifier. Here we choose larger perturbation size $\epsilon = 60$ because we find no obvious pattern with perturbation size $\epsilon = 20$. Observe that clean classifiers have low error rates under test data patched by the alternative triggers. Therefore, clean classifiers are not easily attacked by our approach.

**"Camouflaged" Backdoor** So far we have experimented with triggers that contain colors (i.e., red, blue in Trigger A) that are visually distinctive and as a result, backdoor patterns can be easily recognizable in adversarial examples. We study the case when backdoor trigger is less colorful or contains colors already in the color distribution of clean images. Consider Trigger C in Figure 9a: black and white colors in this trigger are also representative colors of a panda. We train a poisoned binary classifier on ImageNet using Trigger C as the backdoor, where the backdoor attack method is BadNet (Gu et al., 2017). In Figure 9a, we visualize adversarial examples of the *robustified* poisoned classifier. Although there is no clear backdoor pattern in the form of dense color regions, we can observe that in the generated adversarial examples, there is a tendency for black regions to have vertical or horizontal boundaries, which resembles the pattern in Trigger C. Despite the absence of obvious backdoor patterns, we are still able to break the poisoned classifier using cropped patterns from large-$\epsilon$ ($\epsilon = 100$) adversarial examples as shown in Figure 9b. Notice that both of the triggers

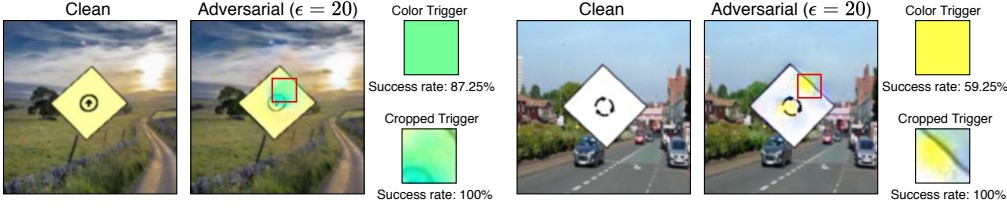

(a) Poisoned Classifier 1  (b) Poisoned Classifier 2

Figure 10: Results of attacking two poisoned classifiers in TrojAI dataset.

| Participants | Denoised Smoothing | | Basic Adv | | Saliency Map |
|---|---|---|---|---|---|
| | participant 1 | participant 2 | participant 3 | participant 4 | participant 5 |
| Accuracy | 94% | 90% | 66% | 82% | 54% |

Table 2: Accuracies that participants obtained for identifying poisoned classifiers in the user study.

are noisy and seem completely different from Trigger C, but they attain higher attack success rate ($88.60\%$ and $83.00\%$) than the original backdoor ($75.80\%$).

## 4.2 TROJAI DATASET

We evaluate our attack on the TrojAI dataset (Majurski, 2020), consisting of a mixed set of clean and poisoned classifiers. TrojAI dataset is initially proposed as a dataset to help develop backdoor detection methods. Here we choose this dataset as it contains a large set of trained poisoned classifiers. Different from ImageNet, we are not aware of the exact backdoor triggers used to poison the classifiers. In Figure 10, we show attack results on two poisoned classifiers. Poisoned classifiers are chosen from Round 0 of the TrojAI dataset. As shown in Figure 10, our methods can attack these poisoned classifiers with high success rate (See Figure 14 in Appendix B for results on more poisoned classifiers.). Similarly, the cropped trigger achieves higher success rate than the color trigger for both classifiers. Especially, notice that both cropped triggers attains $100\%$ attack success rate.

Finally, we conduct a user study on the TrojAI dataset to test the generality of our approach. We develop an interactive tool implementing our method to aid the study. Participants are asked to analyze classifiers with the tool and decide if they are poisoned. Two control groups are used: 1) participants are given a variant of the tool using adversarial examples of the original classifier (denoted as "Basic Adv"); 2) participants are given saliency maps on clean images (denoted as "Saliency Map"). Details on the user study and the interactive tool are in Appendix D. Results are summarized in Table 2, where we show the accuracies of identifying poisoned classifiers for three approaches. Overall, the study suggests that analysts with access to our tool are able to substantially outperform those using alternative methods.

## 5 CONCLUSION

This work shows that backdoor attacks create poisoned classifiers that can be easily attacked even without knowledge of the original backdoor. We find that adversarial examples of a *robustified* poisoned classifier usually contain backdoor patterns. We then construct new poison triggers from the backdoor presence in adversarial examples and show that they give comparable or even better attack performance than the original backdoor.

Our findings urge the research community to rethink the current threat model in backdoor poisoning. It remains to be seen if there exist backdoor attacks that avoid our attack. Our results also raise the question of what is actually learned through the backdoor poisoning process. It seems that backdoor poisoning creates a spectrum of potential backdoors, in addition to the original one. Thus, a rigorous analysis of the implicit effect of backdoor poisoning is needed. More broadly, the idea of *robustifying* (poisoned) classifiers can be a principled approach for analyzing general standard classifiers.

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

# Appendices

## A    EXPERIMENTAL DETAILS

### A.1    TRAINING DETAILS

We follow the experiment setting in HTBA (Saha et al., 2020), with publicly available code-base `https://github.com/UMBCvision/Hidden-Trigger-Backdoor-Attacks`. HTBA divides each class of ImageNet data into three sets: 200 images for generating poisoned data, 800 images for training the classifier and 100 images for testing. The trigger is applied to random locations on clean images. Poisoned datasets are first constructed with corresponding backdoor attack methods. Then we fine-tune the last fully-connected layer of pretrained AlexNet (Krizhevsky et al., 2012) on the created poisoned datasets. The fine-tuning process starts with initial learning rate of 0.001 decayed by 0.1 every 10 epochs and in total takes 10/30 epochs. The number of poisons are 400 images except for BadNet poisoned multi-class classifier, where we find that 1000 poisons are required to achieve high backdoor attack success rate.

We implement the method of CLBD (Turner et al., 2019) utilizing adversarial examples on Ima-geNet. We find that training poisoned classifiers with CLBD is difficult on ImageNet if we follow the exact steps described in Turner et al. (2019). We find that we are able to successfully train poi-soned ResNets (He et al., 2016) by initializing the classifiers with adversarially robust classifiers that are used to generate poisoned data in CLBD. We train adversarially robust classifiers for both binary classification and multi-class classification. For training binary poisoned classifiers, we use 400 adversarial images with perturbation size $\epsilon = 32$ in $l_2$ norm as poisoned data. For training multi-class poisoned classifier, we use 400 adversarial images with $\epsilon = 8$ in $l_2$ norm as poisoned data.

### A.2    COMPUTING ADVERSARIAL EXAMPLE

In our attack, we need to compute adversarial examples of a *smoothed* classifier. To achieve this, we optimize the SMOOTHADV objective (Salman et al., 2019) with *projected gradient descent* (PGD) (Madry et al., 2017; Kurakin et al., 2016). The code for attacking *smoothed* classifier is adopted from public available codebase `https://github.com/Hadisalman/smoothing-adversarial`. Denoiser model is an ImageNet DnCNN (Zhang et al., 2017) de-noiser trained with MSE loss, adopted from the public codebase of *Denoised Smoothing* in `https://github.com/microsoft/denoised-smoothing`.

All adversarial examples are computed by untargeted adversarial attacks with a $l_2$ norm bound $\epsilon$. We use 16 Monte-Carlo noise vectors to estimate gradients of *smoothed* classifiers. The number of PGD steps is 100. Step size at each iteration is $2\times$(perturbation size $\epsilon$) / (# of steps). Except for attacking the poisoned classifier with "camouflaged" backdoor in Figure 9b, where we find that in this case, larger step size leads to slightly better visual results, thus we set step size to be 5 in Figure 9b.

**Deep Dream**    We optimize the adversarial objective with Deep Dream framework adopting the implementation from public codebase `https://github.com/eriklindernoren/PyTorch-Deep-Dream`. We perform 4 iterations, scaling the image by 1.2 every iteration. Due to the large memory requirements of Deep Dream, we use 5 Monte-Carlo noise vectors to estimate gradients. At each iteration, we use 100 steps with step size 5.

**Regularization**    We apply Tikhonov regularization to minimize the $l_2$ norm of image gradients of adversarial perturbations. The image gradient is computed by the filter $F$ in Equation 5. We also experimented with another well-studied denoising objective Total Variation (TV) loss (Rudin et al., 1992), which minimizes the distance between neighboring pixels. TV loss can be seen as a special case of Tikhonov regularization with a specific filter. Comparison of two regularization techniques is shown in Figure 19.

$$F = \begin{bmatrix} 2 & 2 & -1 & -1 \\ 2 & 2 & -1 & -1 \\ -1 & -1 & 0 & 0 \\ -1 & -1 & 0 & 0 \end{bmatrix} \tag{5}$$

## B ADDITIONAL ATTACK RESULTS

### B.1 IMAGENET BINARY POISONED CLASSIFIER

Here we show the complete results for attacking binary poisoned classifiers on ImageNet in Figure 11. Notice that we find effective alternative triggers for all three poisoned classifiers.

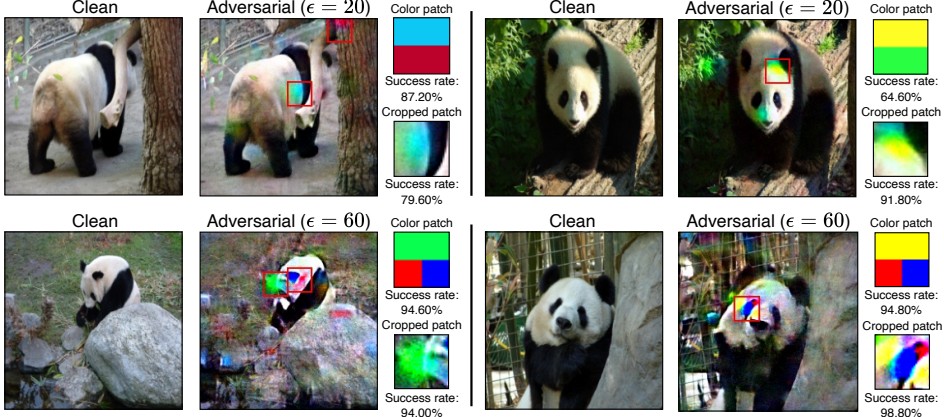

(a) Results for attacking a *robustified* binary poisoned classifier obtained through BadNet (Gu et al., 2017). The attack success rate of the original backdoor Trigger A is 91.60%.

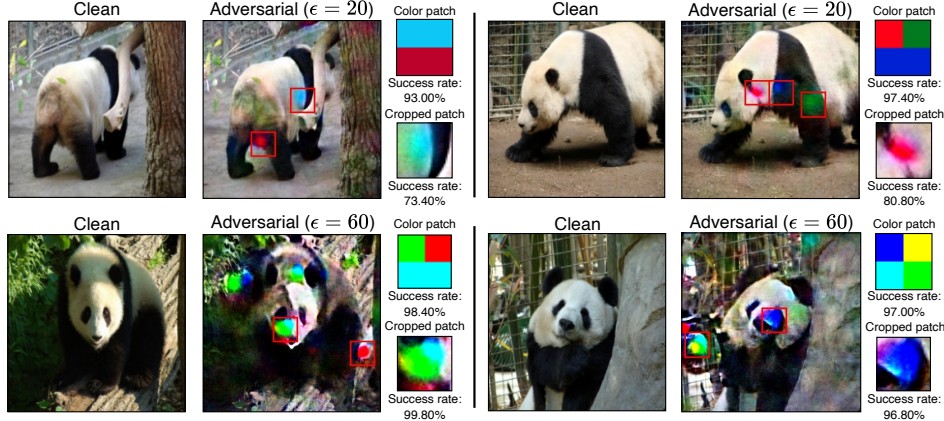

(b) Results for attacking a *robustified* binary poisoned classifier obtained through HTBA (Saha et al., 2020). The attack success rate of the original backdoor Trigger A is 94.00%.

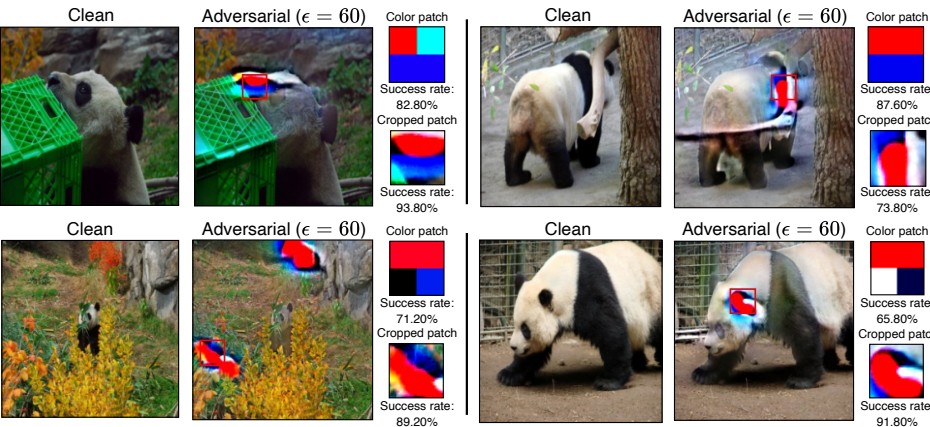

(c) Results for attacking a *robustified* binary poisoned classifier obtained through CLBD (Turner et al., 2019). The attack success rate of the original backdoor Trigger A is 90.00%.

Figure 11: Results for attacking three binary poisoned classifiers obtained by three backdoor attacks.

## B.2 IMAGENET MULTI-CLASS POISONED CLASSIFIER

In Figure 12, we present the results for attacking two poisoned multi-class classifiers on ImageNet obtained by HTBA (Saha et al., 2020) and CLBD (Turner et al., 2019). We can see that our attack constructs effective triggers in both cases.

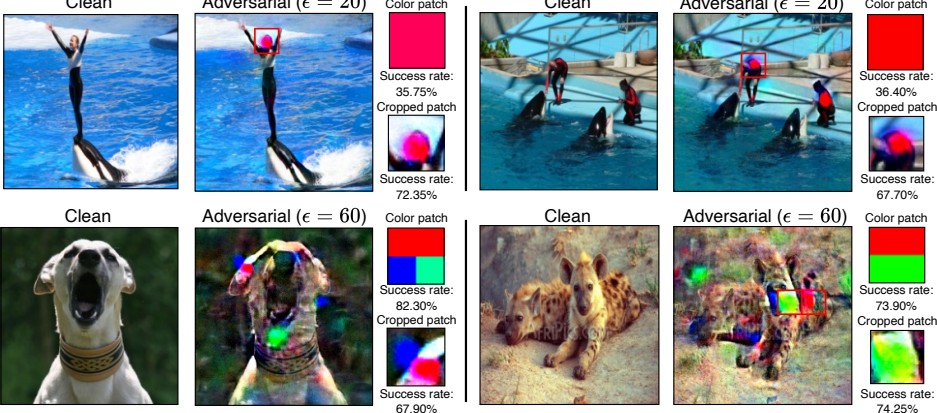

(a) Results for attacking a *robustified* multi-class poisoned classifiers obtained through HTBA (Saha et al., 2020). The attack success rate of the original backdoor Trigger A is $74.55\%$.

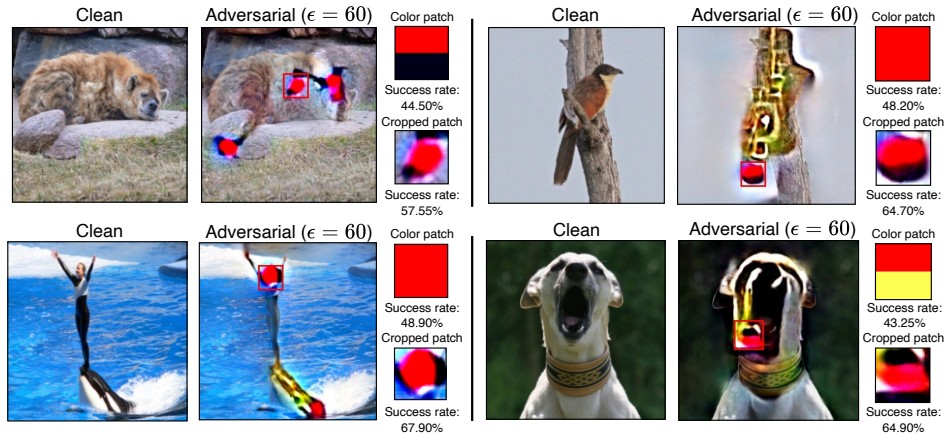

(b) Results for attacking a *robustified* binary poisoned classifiers obtained through CLBD (Turner et al., 2019). The attack success rate of the original backdoor Trigger A is $58.95\%$.

Figure 12: Results for attacking multi-class poisoned classifiers on ImageNet obtained by HTBA (Saha et al., 2020) and CLBD (Turner et al., 2019).

### B.3 IMAGENET BINARY CLEAN CLASSIFIER

In Figure 13, we show the results of attacking a clean binary ImageNet classifier. We can see that the clean classifier is not vulnerable to the triggers constructed by our approach.

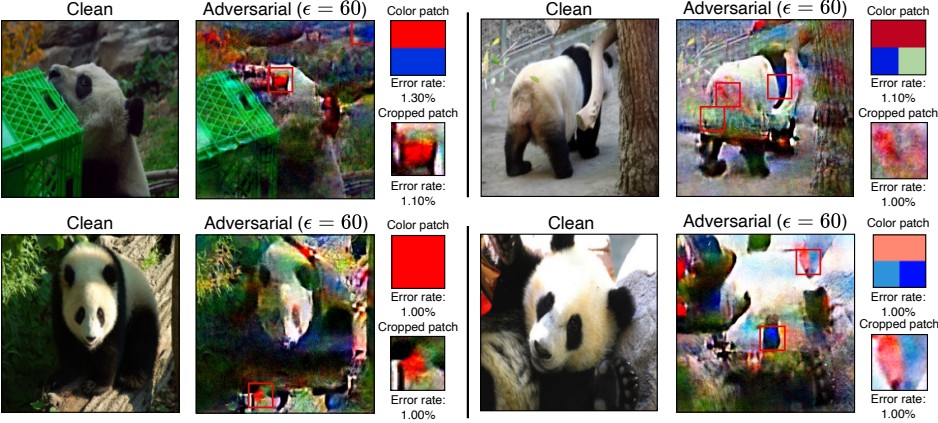

Figure 13: Results of applying our attack on an ImageNet clean classifier (binary).

### B.4 TROJAI

In Figure 14, we show results for attacking poisoned classifiers in the TrojAI dataset. Note that for all 8 poisoned classifiers, the highest attack success rate attained among four alternative triggers is 100%. In Figure 15, we show the results of applying our attack method to two clean classifiers from TrojAI datasets. It can be seen that clean classifiers can classify more than half of the test images correctly even if they are patched by the constructed triggers.

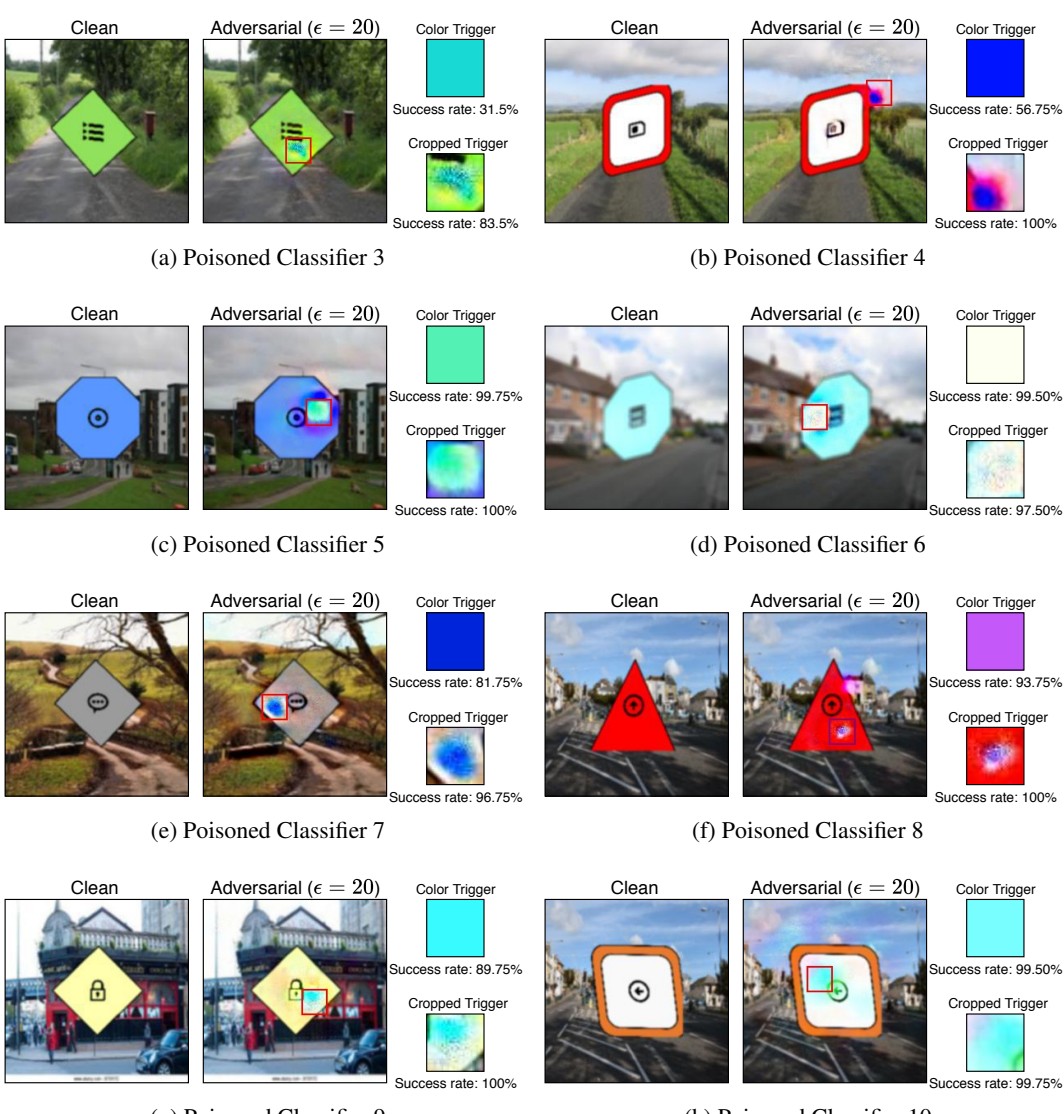

Figure 14: Results of attacking 8 poisoned classifiers in the TrojAI dataset.

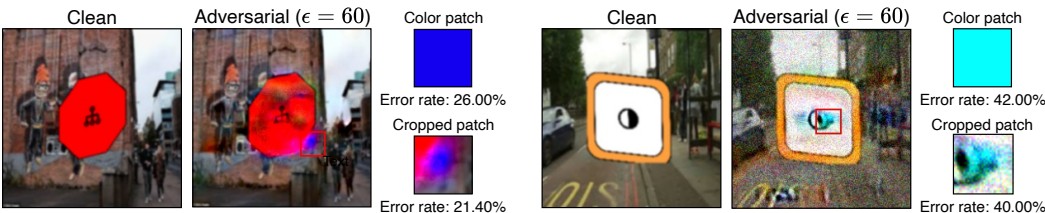

Figure 15: Results of attacking two clean classifiers in the TrojAI dataset.

## C   ADDITIONAL VISUALIZATION RESULTS

### C.1   ADVERSARIAL EXAMPLES ON TROJAI DATASET

Figure 16 presents the adversarial examples of a *robustified* poisoned classifier from the TrojAI dataset, where each row shows images from one class. Below each image we show the class predicted by the poisoned classifier (not the *smoothed* classifier). We highlight those adversarial images with clear backdoor patterns. Note that they are all classified into class 2, which is indeed the target class of backdoor attack. While adversarial images from class 4 (the last row) have dense black regions, we believe that this is a result of mimicking features of class 0 (the class that these images are predicted into) and it can be easily tested using our method that these black regions can not be used to construct successful triggers.

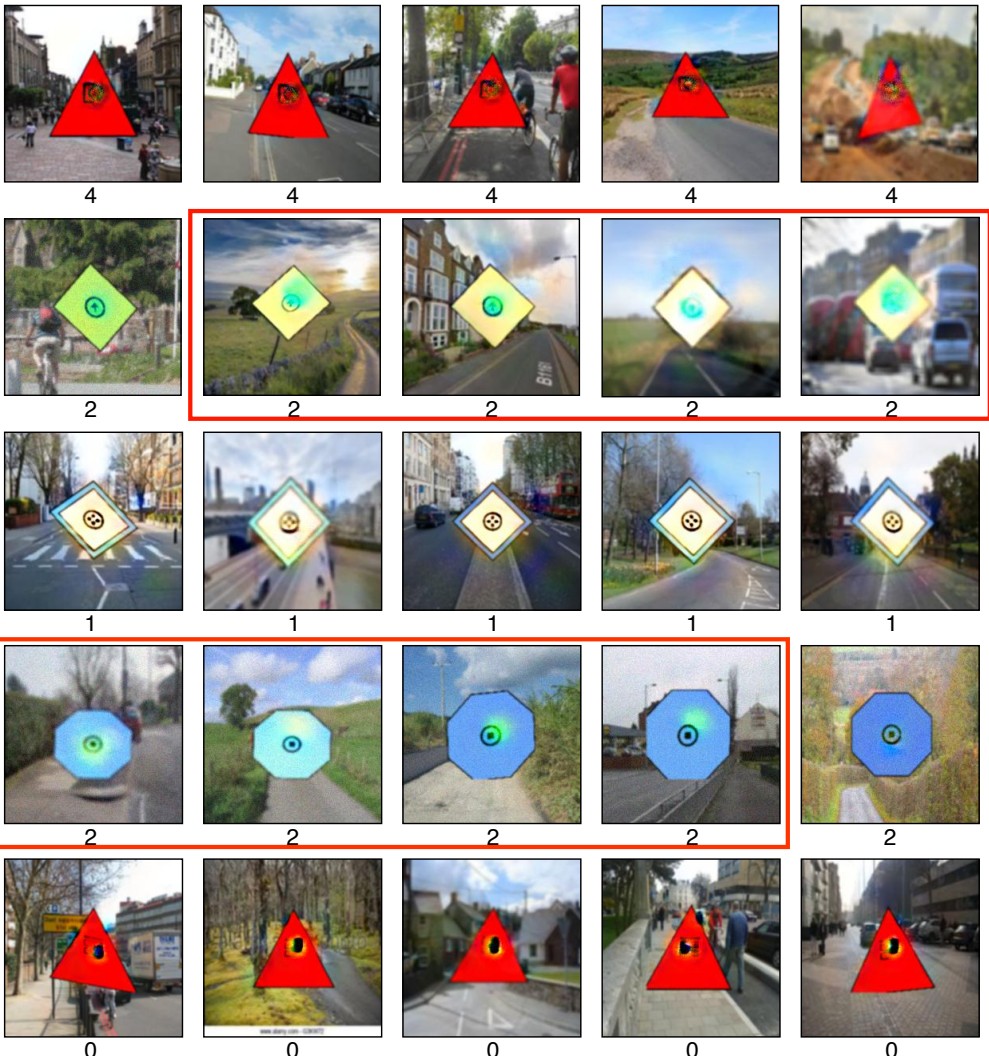

Figure 16: Adversarial examples ($\epsilon = 20$ in $l_2$ norm) of a *robustified* poisoned classifier in the TrojAI dataset. Below each image is the class predicted by the original poisoned classifier.

## C.2 COMPARISON OF DIFFERENT ADVERSARIAL EXAMPLES

Figure 17 shows more results on comparing different adversarial examples ($\epsilon = 20$).

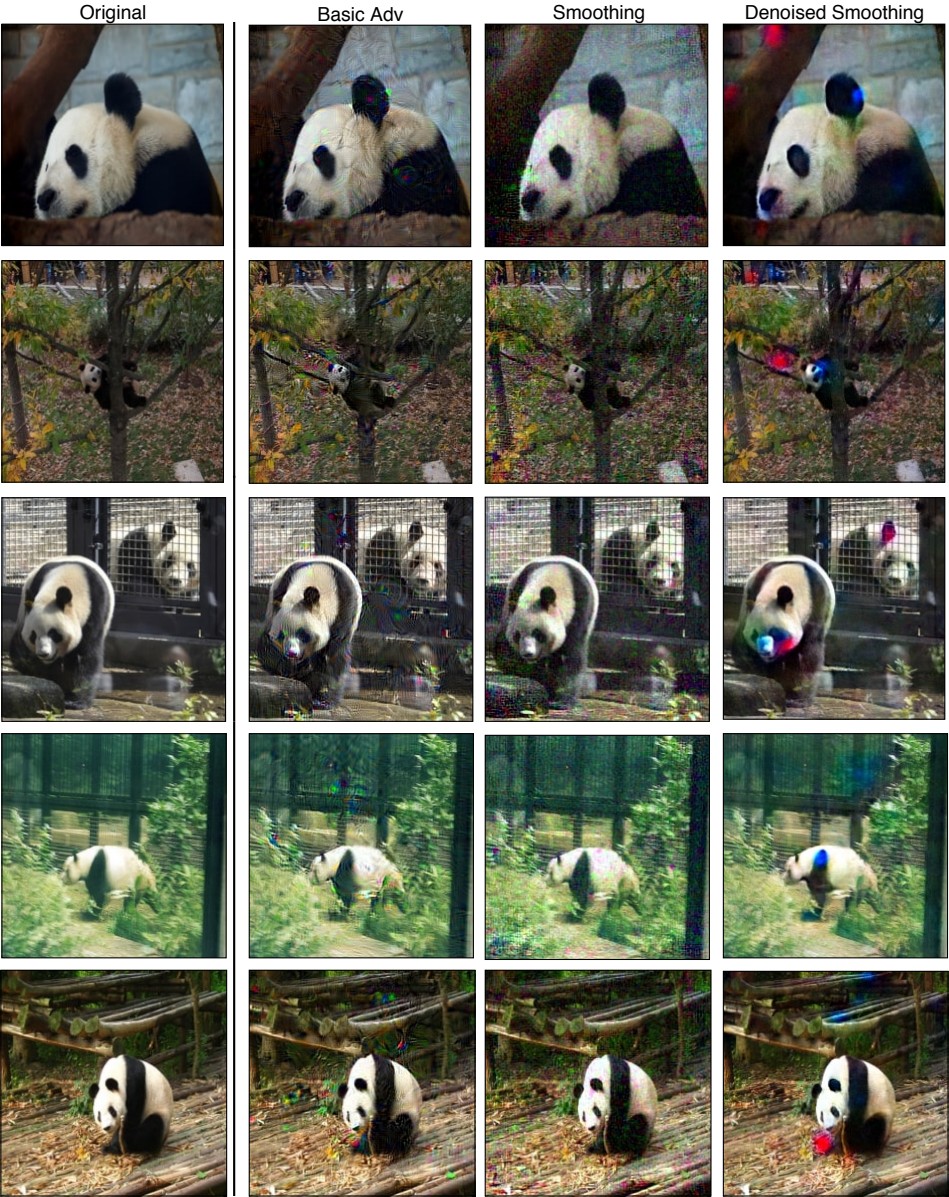

Figure 17: Comparison of different adversarial examples ($\epsilon = 20$) of a *robustified* binary poisoned classifier on ImageNet.

### C.3 ENHANCED VISUALIZATION TECHNIQUES

### C.3.1 DEEP DREAM

Figure 18 shows the comparison of adversarial images with or without enhanced visualization techniques discussed in subsection 3.4. We can see that for Deep Dream, there are more backdoor patterns in a single adversarial image than *Denoised Smoothing*. Together with Tikhonov regularization method, the backdoor patterns become more stable and less noisy.

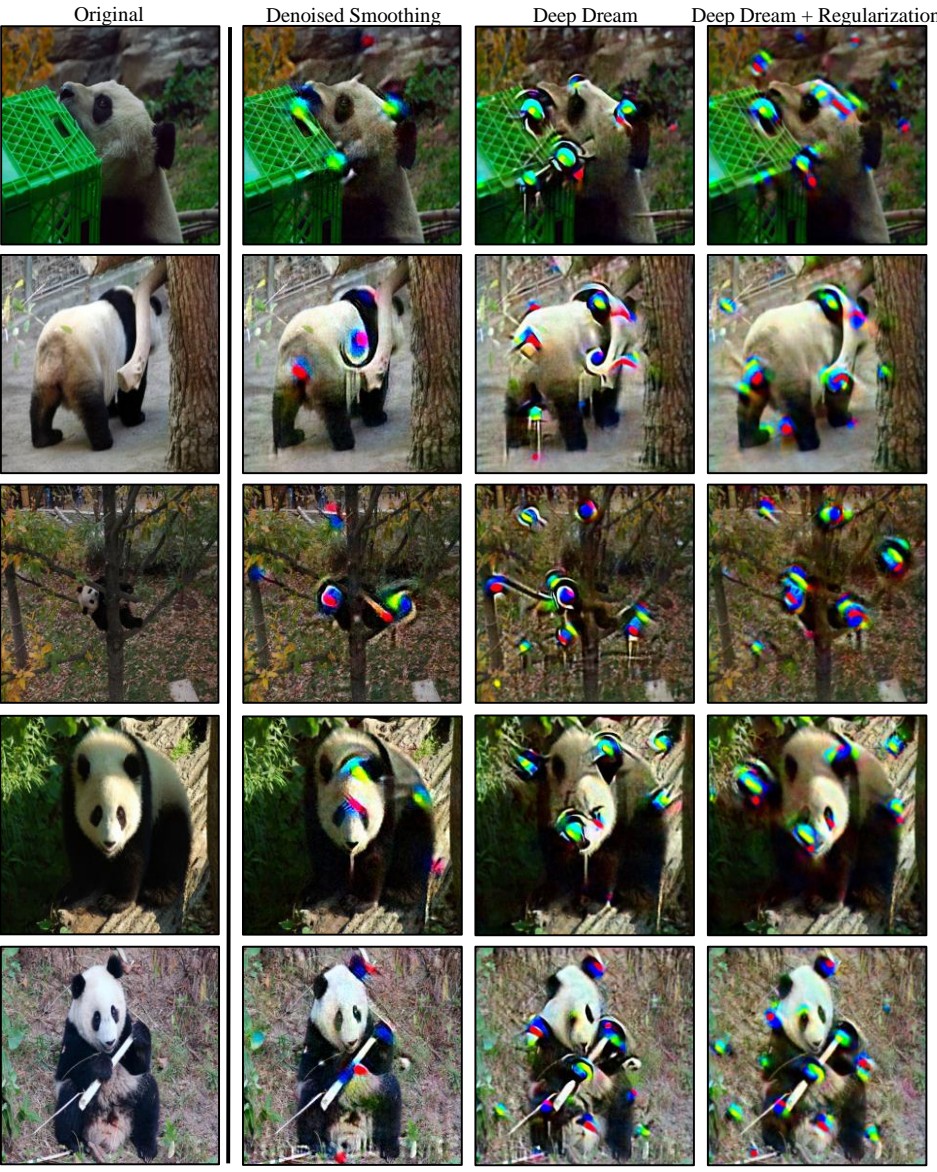

Figure 18: Effects of enhanced visualization techniques on adversarial examples of a *robustified* ImageNet binary poisoned classifier.

### C.3.2 REGULARIZATION

In Figure 19, we show how regularization can be used to reduce background noise in large-$\epsilon$ adversarial examples. We generate adversarial images with $\epsilon = 60$. For *Denoised Smoothing*, we see that there is some background noise. For both regularization techniques, we see that adversarial images are less distorted and there are less noise patterns.

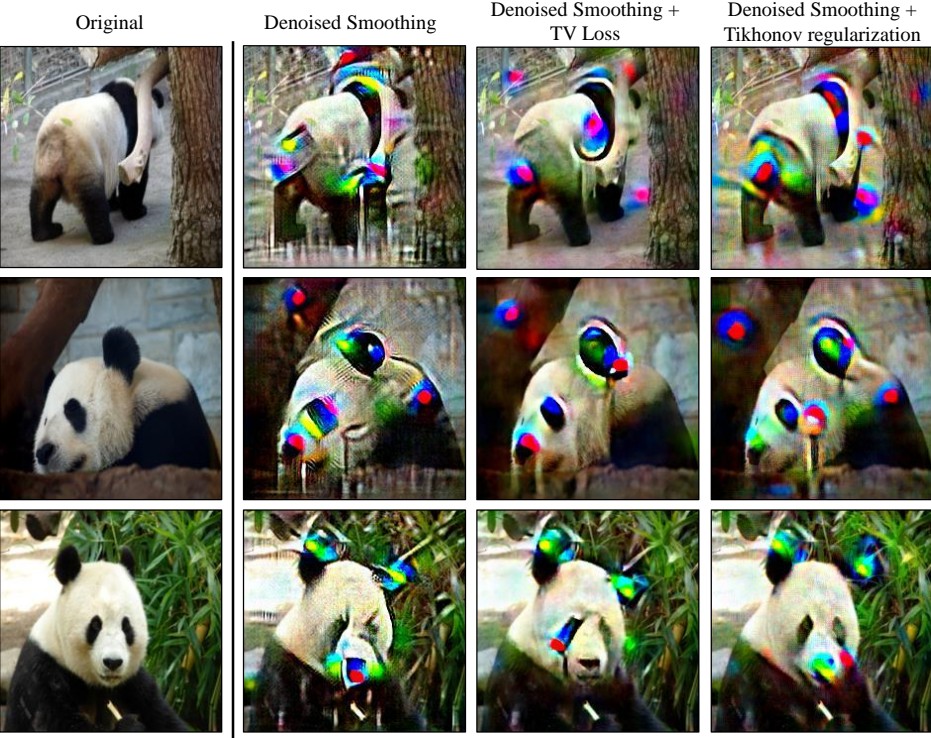

Figure 19: Comparison of adversarial examples generated with/without regularization.

# D    USER STUDY

## D.1    TROJAI INTERACTIVE TOOL

In Figure 20, we show a brief overview of the interactive tool which implements our attack method. The first half of the tool, as shown in Figure 20a, allows users to visualize adversarial examples with chosen attack parameters. Below each image is the class that the adversarial image is predicted. Figure 20b presents the second half of the tool, where users can create new alternative patch triggers and see the classifier's prediction on patched poisoned images.

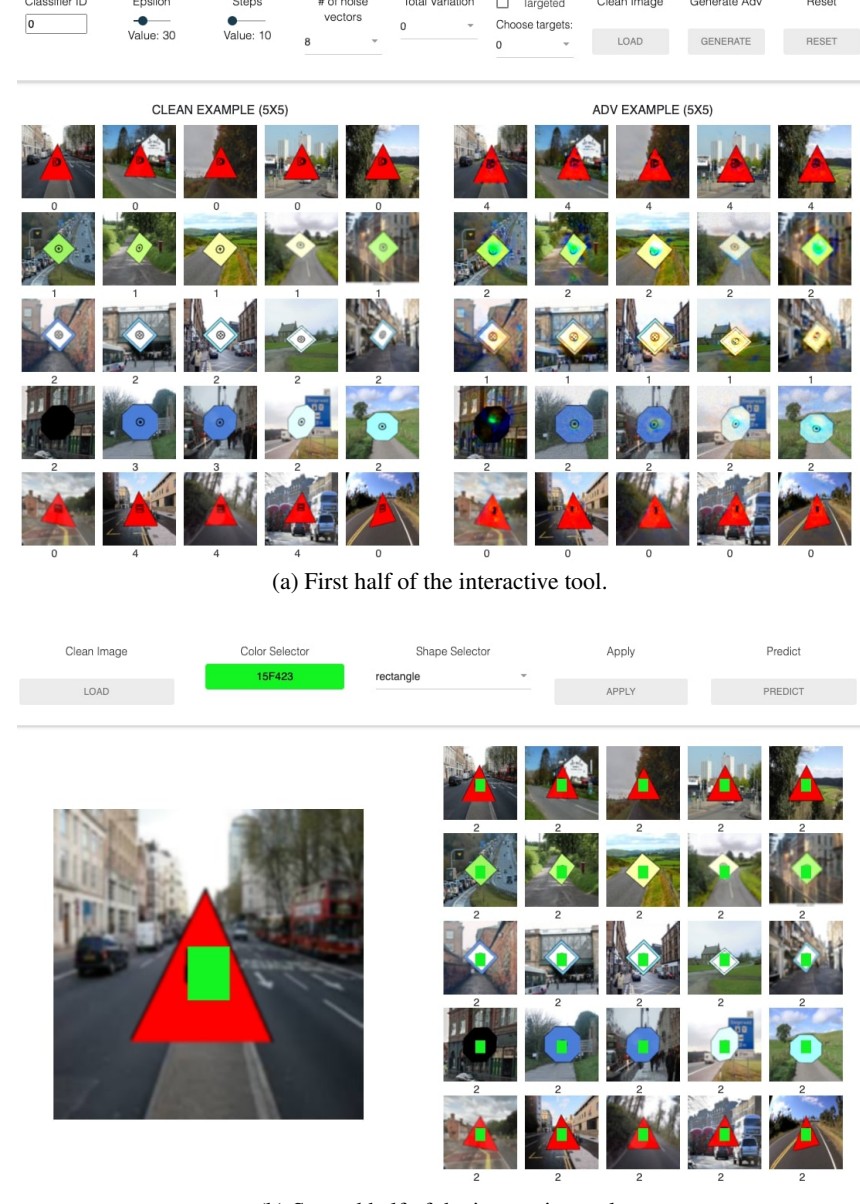

(a) First half of the interactive tool.

(b) Second half of the interactive tool.

Figure 20: Interface of interactive tool we develop for TrojAI dataset.

## D.2 DETAILS ON USER STUDY

We describe our setup for user study in detail. 5 people joined the study. We divide them into three groups: 2 people for *Denoised Smoothing*, 2 people for the control group "Basic Adv" and 1 person for the control group "Saliency Map". For all three groups, participants are asked to mark 50 classifiers as either poisoned or clean. For *Denoised Smoothing* and "Basic Adv", we ask participants to apply our attack method with the interactive tool and test if the model can be successfully attacked by alternative triggers. If so, then mark the classifier as poisoned. For the control group "Saliency Map", Figure 21 shows some sample saliency maps of a poisoned classifier. We use RISE (Petsiuk et al., 2018) to generate saliency maps, as it is shown to outperform other saliency map approaches (Ramprasaath et al., 2017; Marco et al., 2016). For this control group, participants are given the ground-truth labels (poisoned/clean) and saliency maps for 10 classifiers and then try to mark the 50 unlabelled classifiers based on the provided information from 10 labelled classifiers.

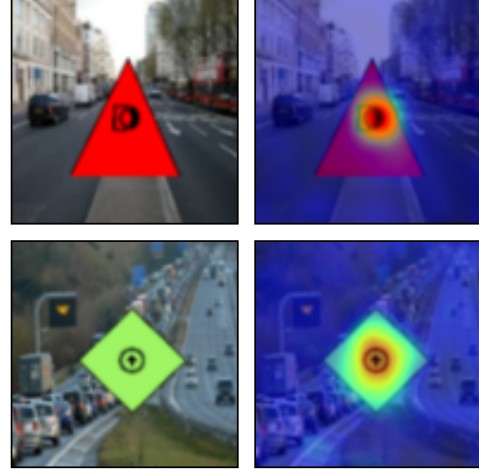

Figure 21: Sample saliency maps of a poisoned classifier on clean images.

# E    THE IMPACT OF TRIGGER LOCATIONS ON BACKDOOR PATTERNS

In this part, we investigate the effect of trigger locations during training on the backdoor patterns in adversarial examples. Specifically, we apply the triggers to fixed image locations (center, lower left, upper left, lower right, upper right ) during training. We use BadNet (Gu et al., 2017) to train poisoned classifiers with Trigger A. Adversarial examples of *robustified* poisoned classifiers are shown in Figure 22. It can be seen that trigger locations do not affect the backdoor patterns in adversarial examples.

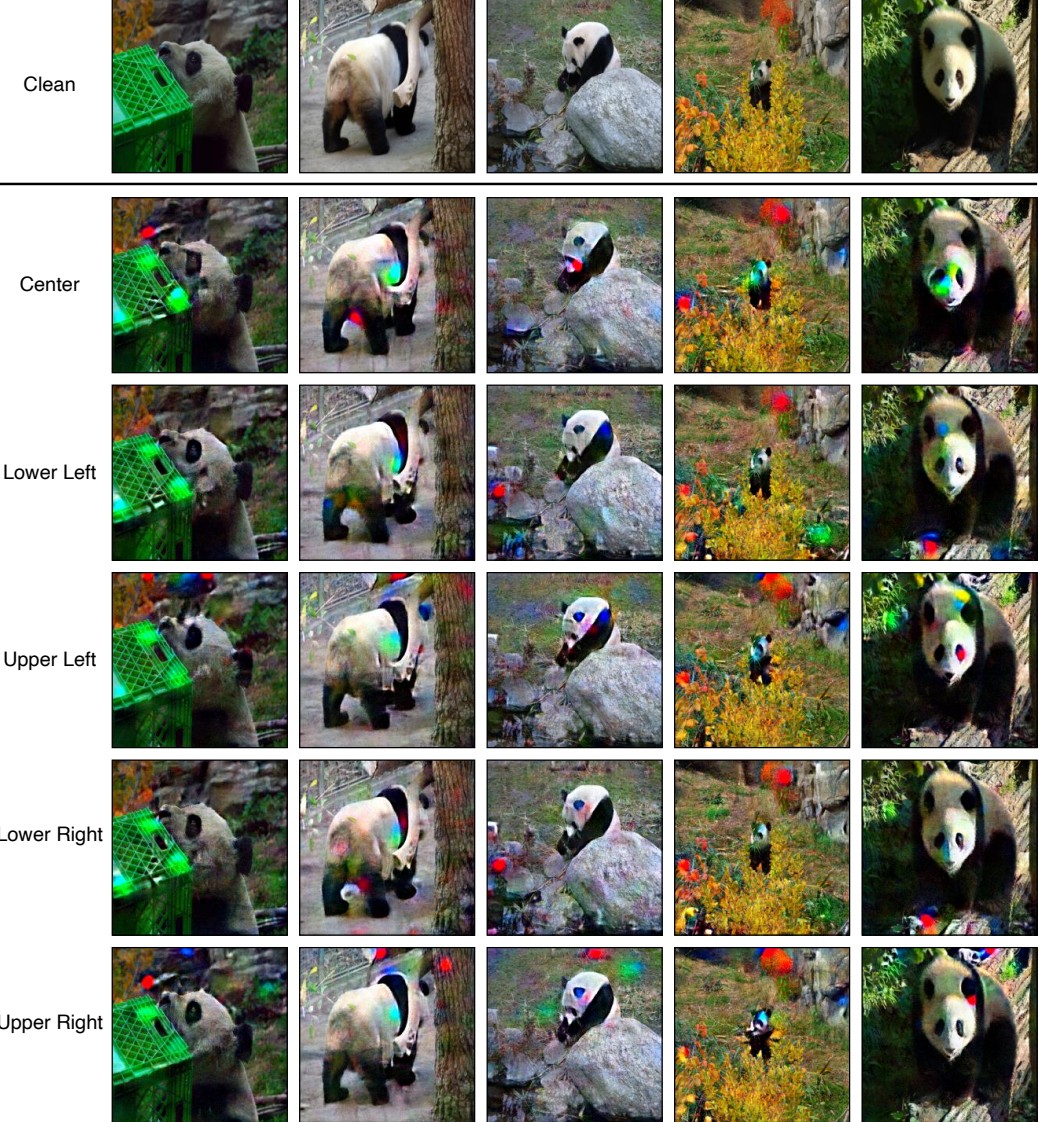

Figure 22: Adversarial examples of *robustified* poisoned classifiers with different fixed trigger locations during training.

# F  IMAGENET CLASSIFIERS WITH MORE CLASSES

In this section, we evaluate our method on ImageNet classifier with more number of classes. We randomly select 10 classes from 1000 ImageNet classes. We then use BadNet (Gu et al., 2017) to train a poisoned classifier with Trigger A. Figure 23 shows the results for attacking this poisoned classifier. We can observe that these alternative triggers have similar or even higher attack success rate than the original trigger.

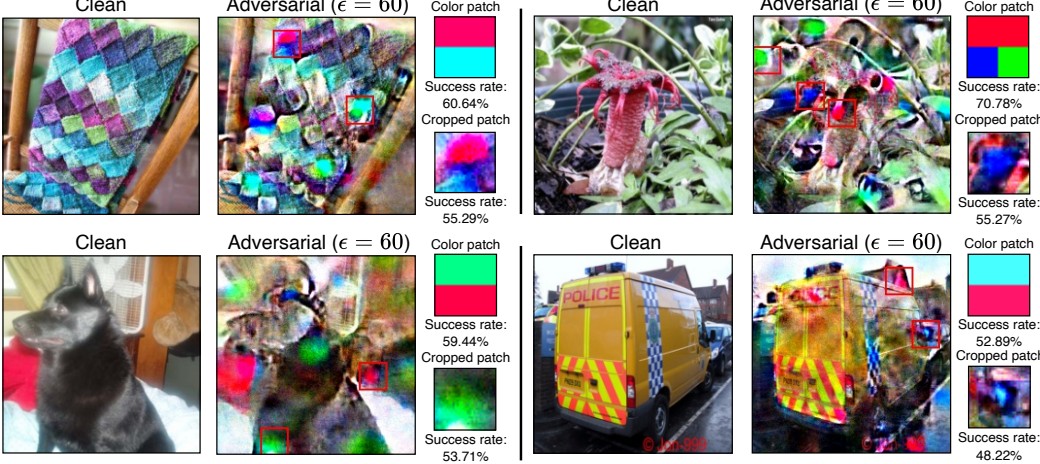

Figure 23: Results of attacking a poisoned ImageNet classifier with 10 classes. The success rate of the original backdoor is 59.71%.

