# OpenReview forum: "Poisoned classifiers are not only backdoored, they are fundamentally broken"
_ICLR.cc/2021/Conference — Reject_

### Official Review · AnonReviewer2 · 2020-10-27
**A expermental report but not a research paper**

**Rating:** 2
**Confidence:** 5

**Review:**

This submission just describes some phenomenons in a strange setting but not proposes any valuable questions.  The authors claim that "anyone with access to the classifier, even without access to any original training data or trigger, can construct several alternative triggers that are as effective or more so at eliciting the target class at test time." However, this paper creates the so-called trigger from the poisoned calssifier. Such a choice is unsual in the adversarial machine larening problem. In most cases, this model is not occupied by the adversary. The authors did not provide some convicing reasons to verify the rationality of this setting. Moreover, the proposed method constructs the alternative triggers by first generating adversarial examples for a smoothed version of the poisoned classifier and then extracting colors or cropped portions of adversarial images. But the motivations of building the  adversarial robust version for the poisoned model and generating the adversarial examples are not well presented in the paper.

---

> ### Author Response · Authors · 2020-11-17
> **Response to AnonReviewer2**
>
> Thank you for your review! We have several clarifications regarding our work:
>
> “A experimental report but not a research paper”
> We do not agree that our paper is not a research paper. First, we propose to use a new method to analyze poisoned classifiers, which we believe is novel. Second, our analysis demonstrates the existence of multiple alternative triggers for a poisoned classifier, which has not been discovered before. We believe our results are valuable for understanding poisoned classifiers. Overall, we believe our work is novel in this research area on backdoor attacks. We would appreciate it if the reviewer can give specifics about why our paper does not count as a research paper.
>
> “This submission just describes some phenomenons in a strange setting but not proposes any valuable questions.”
> We do not believe that our results are in a strange setting. In this paper, we study backdoor attacks, which is a well studied research area [1,2,3,4,5,6]. The settings we investigated in this paper are the common settings for backdoored classifiers. We would appreciate it if the reviewer can point out why our observations are in a strange setting.
>
> “However, this paper creates the so-called trigger from the poisoned classifier. Such a choice is unusual in the adversarial machine learning problem. In most cases, this model is not occupied by the adversary. The authors did not provide some convincing reasons to verify the rationality of this setting.”
> We believe our setting is reasonable. In our method, we assume that we can compute the gradients of poisoned classifiers. We think this is similar to the setting of white-box attacks in related research on adversarial attacks and defense, where white-box attacks also assume one can compute the gradients of victim classifiers directly.
>
> “But the motivations of building the adversarial robust version for the poisoned model and generating the adversarial examples are not well presented in the paper.”
> We think the motivations of our approach are well-presented in the paper. In section 3.1, we have discussed in detail the motivations for building an adversarial robust version of the poisoned classifier. Basically, we want to use the special property of robust classifiers, that they have perceptually-aligned gradients [7]. This property allows us to inspect the adversarial examples of poisoned classifiers in a meaningful way.
>
>
>
> [1] BadNets: Identifying Vulnerabilities in the Machine Learning Model Supply Chain. Gu et al. ArXiv, 2017.
> [2] Clean-Label Backdoor Attacks. Turner et al. https://openreview.net/forum?id=HJg6e2CcK7, 2019.
> [3] Hidden-Trigger Backdoor Attacks. Saha et al. AAAI 2020.
> [4] Clean-Label Backdoor Attacks on Video Recognition Models. Zhao et al. CVPR 2020.
> [5] Trojaning Attack on Neural Networks. Liu et al. NDSS 2018.
> [6] Targeted Backdoor Attacks on Deep Learning Systems Using Data Poisoning. Chen et al. ArXiv 2017.
> [7] Robustness may be at odds with accuracy. Tsipras et al. ICLR 2019.

---

### Official Review · AnonReviewer3 · 2020-10-28
**It is an interesting finding that backdoor-poisoned models are also vulnerable to alternative "triggers", although the reason behind it is unknown**

**Rating:** 5
**Confidence:** 3

**Review:**

Summary:
This paper demonstrates that backdoor-poisoned machine learning models can also be vulnerable to alternative triggers. Specifically, adversarial samples that are generated against models robustified with Denoised Smoothing often show backdoor patterns. Therefore, these adversarial samples can be used to create new triggers either by (1) choosing a representative colour from a backdoor pattern, or (2) cropping an image that contains a backdoor pattern. Experimental results suggest that alternative triggers can be equally or even more effective than the original trigger.

Pros:
1. This paper studies an important question of the vulnerability of backdoor-poisoned machine learning models.
2. Two methods are proposed to generate alternative triggers that can cause the poisoned machine learning model to misbehave.

Cons:
While it is a novel finding that poisoned machine learning models are not only vulnerable to the initial triggers, I have the following questions on the proposed method:

1. The biggest concern is that as pointed out in Section 3.3, both the two proposed strategies for generating alternative triggers require human inspection. Can these triggers be generated automatically? How easy is it for an algorithm to identify the backdoor pattern, and then to crop a patch image that contains the pattern?
2. What could be the potential reason why the proposed approaches are effective for generating alternative triggers? What could be the relation between Denoised Smoothing and the backdoor pattern? Could there be other ways to create triggers? There is a lack of discussion on this issue.
3. In terms of parameters, the perturbation size \epsilon is set to 20 and 60 (in l2 norm) in the experiments. What is the impact of \epsilon on the attack success rate? Specifically, with a smaller value of \epsilon, can the attack still achieve such high success rate?

---

> ### Author Response · Authors · 2020-11-17
> **Response to AnonReviewer3 [2/2]**
>
> ** Perturbation size **
> From the observations we made from Figure 7, Figure 11 and Figure 12, in general, epsilon 60 leads to better attack success rate than epsilon 20. For perturbation size smaller than 20, we do not expect it to have better attack success rates because if the perturbation size is small, the backdoor patterns do not occur or they are not obvious as compared to larger perturbation size. This is also the reason why we choose 20 as the minimum perturbation size to experiment with.
>
> Thank you for your review again! Any further questions or suggestions are welcome.
>
> [1] Robustness may be at odds with accuracy. Tsipras et al. ICLR 2019.
> [2] Image Synthesis with a Single (Robust) Classifier. Santurkar et al. Neurips 2019.
> [3] Are Perceptually-Aligned Gradients a General Property of Robust Classifiers? Kaur et al. ArXiv 2019.
> [4] Defense against adversarial attacks using high-level representation guided denoiser. Liao et al. CVPR 2018.
> [5] Mitigating Adversarial Effects Through Randomization. Xie et al. ICLR 2018.
> [6] PixelDefend: Leveraging Generative Models to Understand and Defend against Adversarial Examples. Song et al. ICLR 2018.
> [7] Obfuscated Gradients Give a False Sense of Security: Circumventing Defenses to Adversarial Examples. Athalye et al. ICML 2018.

---

> ### Author Response · Authors · 2020-11-17
> **Response to AnonReviewer3 [1/2]**
>
> Thank you for your detailed and valuable review. We are very happy to address your concerns.
>
> ** Human inspection **
> First, we do agree that our procedure for generating alternative triggers requires human inspection. However, we believe these efforts are minimal and easy. As discussed in the paper, both image cropping and pixel selection are very basic operations. Both operations can be achieved in python in a few lines of code, which can be seen in the anonymized code we have uploaded to the supplementary material in the recent version.
>
> Second, our work follows the line of previous work [1,2,3] on visualizing adversarial examples of robust classifiers. [1,2,3] shows the property of perceptually-aligned gradients through human evaluation of adversarial examples. Therefore, we believe the dependency of our method on human inspection is reasonable.
>
> Last, we view the most important contribution of our work as discovering the existence of multiple alternative triggers, which is the message we want to highlight in this paper. The approach we use to construct these triggers is a means to an end. There may be other ways of constructing these triggers, but we think for this paper, this new property of poisoned classifiers is a more important contribution.
>
> ** Automatic generation of alternative triggers **
> In fact, during the development of this work, we have made some attempts at automatically generating color patches. Specifically, we achieved this by three steps:
> 1. Extracting the pixel locations where the difference of adversarial examples and clean images examples is greater than a threshold. This step is to potentially extract the regions of backdoor patterns.
> 2. Then plot the HSL/HSV color histogram of these pixels. The reason we use HSL/HSV color histogram is because we can use the hue channel to analyze which colors appear in backdoor patterns (as compared to RGB, where three channels are required to represent a color).
> 3.  Compare the color histograms of both adversarial examples and clean images. We can identify the colors prominent in backdoor patterns.
>
> We found that this approach is able to successfully extract the colors in backdoor patterns. As for the cropped patch, we did not try an algorithmic approach before. However, we think it is possible. We can use the algorithm to extract colors to find those pixel locations for the extracted colors, then we can crop the regions around these pixels as cropped patches.
>
> The reason we did not put it in the paper is we believe that we do not want to shift the focus of our paper. The main point we want to make in the paper is the existence of multiple alternative triggers for poisoned classifiers, which we believe is valuable and important for this research area. Therefore, for simplicity, we did not include our color histogram extraction algorithm in the paper.
>
> ** What could be the potential reason why the proposed approaches are effective for generating alternative triggers? What could be the relation between Denoised Smoothing and the backdoor pattern? Could there be other ways to create triggers? **
> We believe that the reason why our approach works is straightforward. As shown in [1,2], adversarial examples of robust classifiers resemble instances from the misclassified classes. In backdoored classifiers, the poisoned examples are classified into the target class. Thus the poisoned examples can be seen as instances of the target class. Then if we generate adversarial examples of robustified poisoned classifiers towards the target class, we expect the adversarial examples to mimic poisoned examples. In fact, our experimental analysis (see Figure 2 and Figure 3), backdoor patterns do exist in adversarial examples. Given the backdoor patterns in adversarial examples, it is then natural to test how well these backdoor patterns perform in terms of attack success rate, which is what we do in the paper.
>
> As for Denoised Smoothing, in this paper, Denoised Smoothing is merely an approach we use to convert poisoned classifiers into robust classifiers, of which we can generate perceptually aligned gradients. There are some works [4, 5, 6] that also propose to convert pre-trained classifiers into robust ones, however, they have been shown to be vulnerable to adaptive attacks [7]. Different from these empirical defenses, Denoised Smoothing is a certified defense, therefore the robustified classifiers are guaranteed to be robust.
>
> We do think that there could be other ways to create triggers. However, we think using Denoised Smoothing is a well-motivated approach with nice visual examples. Moreover, we think the most important contribution of our work is not the method we use for analysis, but the observations we made on poisoned classifiers that multiple alternative triggers exist. We believe that this property is useful and necessary for understanding backdoor attacks.

---

> > ### Comment · AnonReviewer3 · 2020-11-25
> > **Further on Question II**
> >
> > Thank you for the reply.
> >
> > Regarding the second question on the potential reason, it is mentioned in Section 3.3 that the proposed attack is only effective against backdoored classifiers but not clean classifiers. However, following your explanation, shouldn't clean classifiers be vulnerable as well?

---

### Official Review · AnonReviewer4 · 2020-10-28
**Interesting study,  some concerns regarding the method and experiments**

**Rating:** 5
**Confidence:** 3

**Review:**

Summary:
This is an interesting study on the analysis of poisoned classifiers and backdoor attacks. The authors showed that with some post-processing analysis on a poisoned classifier, it is possible to construct effective alternative triggers against a backdoor classifier. In particular, after creating several poisoned classifiers, and smoothing them using a Denoised smoothing technique, one can generate adversarial examples. Using these examples, it is possible to extract color or cropped patch as new triggers to break the poisoned classifier.

Reason for score:
The process of generating alternative triggers and finding effective triggers for each backdoor attacker is mainly manual and needs human intervention. This makes the proposed solution very challenging. Also, the experimental results do not clearly show the generalizability of the model, especially on more difficult backdoor attacks. I think more experiments need to be done to evaluate the consistency of the results and to study the generalizability of this work across various datasets.


More detailed comments:
-         How the adversarial examples of robustified poisoned classifiers look like when we have invisible backdoor attacks? This is important since the trigger generation is directly related to the backdoor pattern of the generated adversarial examples. Did the authors investigate this matter?
-         Does changing the location or appearance of the original trigger affect the backdoor patterns in adversarial examples?
-         In the experiment section, it would be better if the authors showed both color patch and cropped patch results on fixed images and shows which technique has a higher success rate.
-         In the experiment section, the authors only showed two samples to prove that clean classifiers are not easily broken. Please show the overall success rate for clean classifiers on both datasets as well.
-         It would be better if the authors also showed the results on larger datasets with more number of classes.
-         I would recommend to open source the code.

Minor comments:
-         In section 3, please briefly explain how the poisoned classifier is trained using the two triggers (it was not clear before reading the experiment section)
-         For table 1, please mention the number of samples used to calculate the success rate. The authors mentioned the highest success rate is picked for different triggers. It would be great to see how each trigger performs separately regarding the success rate.
-         Some references are missing, please add more recent papers. For example:
Rethinking the Trigger of Backdoor Attack
Invisible Backdoor Attacks on Deep Neural Networks via Steganography and Regularization
etc.

---

> ### Author Response · Authors · 2020-11-17
> **Response to AnonReviewer4 [2/2]**
>
> ** Location and appearance of trigger **
> We have trained BadNet poisoned classifiers with different trigger locations (center, upper left, upper right, lower left, lower right). We plotted the adversarial examples of robustified poisoned classifiers. We have added the results in Figure 20 in Appendix E in the new version. We find that the trigger location during training does not affect the backdoor patterns in adversarial examples.
>
> The appearance of trigger does affect the backdoor patterns. In Figure 3, we use two backdoors (Trigger A and Trigger B) and we can see that the backdoor patterns are different. For example, adversarial examples of robustified poisoned classifier 2 (Trigger B) only have special pink regions, which is different from Trigger A. We also have an additional analysis in the paper on the effect of trigger color on backdoor patterns. Results are in Figure 4, where we find that different triggers lead to different backdoor patterns in adversarial examples.
>
> ** Color and Cropped patch on fixed images **
> We have updated the experiment section to address this issue. Specifically, for each adversarial example, we show both color patch and cropped patch constructed from this example. We have updated Figure 7, 11 and 12 accordingly. In general, we find that whether color patch or cropped patch perform better depends on the example considered. Last, we would like to mention that for five of six poisoned classifiers we experimented with, the highest attack success rates in Table 1 are achieved by cropped patches. This may suggest that cropped patches may be more effective overall. We have included these observations and discussions in the paper.
>
> ** Results for clean classifiers and more number of classes **
> Thanks a lot for the suggestions. We will add more results on clean classifiers and ImageNet classifiers with more number of classes in the paper soon.
>
> ** Source code **
> We have uploaded our anonymized code in the supplementary material in the new version.
>
> ** Minor comments **
> We have updated our paper to address the issues the reviewer mentioned. Our modifications are as follows:
> 1. In section 3, we added a sentence describing how we obtained the backdoor poisoned classifiers.
> 2. In section 4, we included the number of test samples we used to compute the attack success rate. Specifically, on ImageNet, we used 50 images per source class, which is 50 images for the binary classifier and 200 images for the multi-class classifier. On TrojAI dataset, we used the 500 sample test images provided along with each classifier. In the updated version, we added this information in the first paragraph of section 4.
> In terms of how each trigger performs separately regarding the success rate, we have already included all of it in the initial version. In Figure 7, we showed the four triggers for BadNet poisoned multi-class classifier, which corresponds to one entry in Table 1. In Figure 11 and Figure 12 in Appendix B, we presented the results for other five entries in Table 1. We have updated the descriptions in the paper to resolve the confusion and make this point more clear.
> 3. We have added the references of two papers in the related work part.
>
> Thank you for your review again! Any further questions or suggestions are welcome.
>
> [1] BadNets: Identifying Vulnerabilities in the Machine Learning Model Supply Chain. Gu et al. ArXiv, 2017.
> [2] Clean-Label Backdoor Attacks. Turner et al. https://openreview.net/forum?id=HJg6e2CcK7, 2019.
> [3] Hidden-Trigger Backdoor Attacks. Saha et al. AAAI 2020.
> [4] Clean-Label Backdoor Attacks on Video Recognition Models. Zhao et al. CVPR 2020.
> [5] Trojaning Attack on Neural Networks. Liu et al. NDSS 2018.
> [6] Neural Cleanse: Identifying and Mitigating Backdoor Attacks in Neural Networks. Wang et al. S&P 2019.
> [7] Exposing backdoors in robust machine learning models. Soremekun et al. ArXiv, 2020.
> [8] Practical detection of trojan neural networks: Data-limited and data-free cases. Wang et al. ECCV 2020.
> [9] Tabor: A highly accurate approach to inspecting and restoring trojan backdoors in ai systems. Guo et al. ICDM, 2020.
> [10] Detecting Backdoor Attacks on Deep Neural Networks by Activation Clustering. Chen et al. ArXiv, 2018.

---

> ### Author Response · Authors · 2020-11-17
> **Response to AnonReviewer4 [1/2]**
>
> Thank you for your constructive and valuable feedback. We are very happy to address your concerns.
>
> ** Human inspection **
> Indeed, we propose an attack procedure that relies on human evaluation of adversarial examples. However, we believe that the major contribution of this work is that we demonstrated the existence of these alternative triggers that are as effective or more so than the backdoor. We do agree that our approach is an important part of the paper. But the central message we want to convey in this paper is this special property of backdoor poisoned classifiers which have not been discovered before and we hope this new discovery can help better understanding of  poisoned classifiers.
>
> Second, the human intervention in our approach is not huge, for two reasons: 1) the backdoor patterns are easy to identify by human eyes, which can be seen by the adversarial examples provided in the paper. Also in the paper we have a human study (Table 2) which supports this argument; 2) both image cropping and selecting color pixels are very simple. Both operations can be done in a few lines of python code, which can be seen in our source code uploaded in the supplementary material.
>
> Last, during the development of this work, we have made some preliminary attempts at automatically extracting the alternative triggers. We considered extracting the dominant colors in the backdoor patterns by color histogram analysis. We had some success at automatically extracting the colors in backdoor patterns and reconstructing the color patch. However, we find it more important to highlight the discovery we made on backdoor poisoned classifiers. Therefore, for simplicity, we did not include it in the paper.
>
> ** Generalizability across various datasets **
> We believe the current datasets we use in the paper (ImageNet and TrojAI datasets) are enough to show the generalizability of our approach. ImageNet is the standard large-scale vision dataset nowadays to evaluate computer vision and machine learning tasks. TrojAI is a more domain-specific dataset, created specially for studying backdoored classifiers. TrojAI dataset consists of various model architectures (ResNet, Inception, DenseNet, etc) and backdoor patterns. Also, both datasets have high resolution images (224x224x3 inputs after preprocessing). Therefore we believe our methods are both generalizable and practical.
>
> ** Invisible backdoor attacks **
> First, we would like to clarify in this paper, we study the most common and widely studied setting of backdoor attack [1,2,3,4,5]. In this setting, the backdoor trigger is an image patch and applied to clean images by replacing part of the clean images. A significant number of backdoor defense methods is also developed based on this backdoor attack setting [6,7,8,9,10]. Our work is also focused on backdoored classifiers in this setting. Therefore, we believe that showing the existence of multiple alternative triggers for such backdoored classifiers is a major contribution in this research area.
>
> Second, for invisible backdoor attacks, for instance, in the backdoor attack proposed in the paper “Invisible Backdoor Attacks on Deep Neural Networks via Steganography and Regularization”, the backdoor trigger takes a different form. In that paper, steganography based backdoor modifies the LSB bits of pixels and regularization based backdoor is a global noise pattern. Since our approach is not developed based on these advanced backdoor formulations, we do not expect our approach to work in this case. However, the observation we made from traditional backdoored classifiers could hold in this case: there may exist multiple alternative backdoor triggers for these invisible backdoor attacks, with the only difference that the backdoor formulation is different. It would be interesting future work to see if there are backdoor attacks that do not lead to multiple alternative triggers.
>
> Last, we did investigate a type of “invisible” backdoor attack in the paper. We consider the setting where the image patch is camouflaged. In other words, the image patch consists of colors existing in clean images (see Trigger C in Figure 9a). This can be seen as an example of  invisible backdoor as the camouflaged trigger blends in well with the clean images. In the paper, we find that our approach is even effective in this case (details in last paragraph of section 4.1).

---

> ### Author Response · Authors · 2020-11-19
> **Updates on experiments**
>
> We have added the results for attacking clean classifiers and also classifiers with more number of classes. Details are as follows:
>
> ** Clean classifiers **
> For more samples on clean classifiers, we have added more results in the new version. Specifically, in Figure 8, we show two more samples of attacking the clean classifier. Also, in Figure 8, we include the results of both color patch and cropped patch. For the TrojAI dataset, we have added the attack results on two clean classifiers in Figure 15 in Appendix B.4.
>
> There is one thing we would like to clarify regarding the results on clean classifiers. For clean classifiers, there is no target class. Thus the metric attack success rate is not valid for clean classifiers. Therefore, we use the error rate of clean classifiers on test data patched with the constructed alternative triggers. We use this metric to evaluate the vulnerability of clean classifiers to the alternative triggers. If the error rate is low, it means that clean classifiers are not susceptible to the constructed triggers. We are sorry that we did not define this in detail in the initial version. We have made this point clear in the new version and also in Figure 8, we use the term “error rate” instead of “success rate”.
>
> ** Classifiers with more number of classes **
> We have trained a poisoned ImageNet classifier with 10 classes. We used BadNed for training the poisoned classifier. The ten classes are randomly selected from 1000 ImageNet classes. We apply our approach on this poisoned classifier and we find that the alternative triggers we construct have similar or even higher attack success rate than the original trigger. The detailed results are in Figure 23 in Appendix F.
>
> Thank you again for your suggestions. If the reviewer has more questions on the additional experiment results, we are happy to answer.

---

### Official Review · AnonReviewer1 · 2020-10-28
**introducing backdoor triggers to classifiers makes them vulnerable to alternative attacks, not just those intended via the original trigger.**

**Rating:** 7
**Confidence:** 4

**Review:**

## Summary ##
Suppose that we wish to have a classifier that works well under normal circumstances, but fails when we want it to. One way to do so is via data poisoning attacks: introduce a special datapoint to the training data, such that if the classifier observes it, all of its labels become completely broken.

The basic premise of the paper is that poisoned classifiers are broken in a fundamental way - not only are they vulnerable to attacks based on the original trigger image, they are also vulnerable to attacks by adversaries who do not know the original trigger. This is interesting for both the learning and the privacy communities, as it shows that backdoor attacks introduce a host of vulnerabilities far beyond that which was assumed.

To test this hypothesis, the authors show that one can design novel triggers on poisoned classifiers, that, in some cases, outperform the attacks based upon the original trigger.

Similar avenues do not work as effectively on “clean” classifiers, where no triggers have been introduced.

The authors show that their methodology is effective on a variety of datasets; in fact, the authors show that users are better able to distinguish poisoned classifiers even without the backdoor present.

Overall I liked the authors’ approach, and the paper was a pleasure to read. The authors make clear, concise and refutable claims, and proceed to analyze them in a rigorous manner.

## Pros ##
1. The paper tackles an interesting and well-motivated problem, and shows that a single line of attack is much more worrisome than previously thought.
2. The paper is well written, all claims are easy to follow.
3. I like the fact that the authors incorporated user studies into the evaluation.

## Cons ##
1. It is not fully clear whether the approach extends beyond theoretical interest and curiosity. Is there a clear and immediate implication for how we train or protect our ML models?
2. The paper provides a heuristic approach that may or may not be generalized beyond the datasets that were tested. There is no analysis showing why the approach makes sense.

## Questions to the Authors ##
1. How does this line of work relate to formal privacy notions? The authors mention a few similar modes of attack, but am I correct in assuming that differentially private training methods are immune to such attacks?
2. If we use robust training methods, does this issue go away?

## After Rebuttal ##

I thank the authors for their response to my question. I think that the comment regarding differential privacy being ineffective is particularly interesting. It would be nice to actually demonstrate this empirically - construct a DP classifier with a poisoned backdoor, and show that the authors' method is still effective.

My support for the paper remains unchanged.

---

> ### Author Response · Authors · 2020-11-17
> **Response to AnonReviewer1**
>
> Thank you for your detailed review. We are happy to address your concerns.
>
> ** Extension and Implication **
> A direct implication of our results is that we need to rethink backdoored classifiers. On one hand, for an adversary who aims to poison a classifier, he/she should not expect that the embedded backdoor trigger which he/she holds is secret. Instead, the adversary should be aware of the possibility that it is easy for other people to attack the backdoored classifiers in the same way he/she does. On the other hand, for machine learning practitioners who train and deploy their models, our results suggest that they need to be more careful on sanitizing their data. Otherwise, backdoored classifiers could pose much greater danger than previously thought.
>
> ** Generalization beyond datasets tested **
> We evaluated our attack on two large scale datasets: ImageNet and TrojAI. Both datasets consist of high resolution images (224x224). ImageNet is the de facto dataset for evaluating common vision tasks. TrojAI dataset is specially created to investigate backdoor poisoned classifiers, with a huge amount of pretrained poisoned classifiers. TrojAI dataset contains a wide variety of model architectures (ResNet, Inception, DenseNet, etc) and backdoor patterns. Therefore, we believe these two datasets are enough to show the generality of our attack.
>
> ** Why our approach makes sense **
> In terms of why backdoor patterns exist in adversarial examples, we believe we have discussed it in section 3.1 in detail. The reason is that we are able to construct a robustified poisoned classifier (with Denoised Smoothing). Robust classifiers have perceptual-aligned gradients [1], which basically says that adversarial examples of robust classifiers resemble instances from the misclassified class. In the setting of backdoor poisoned classifiers, the backdoored instances B(x) can be seen as instances of the target class, then if we generate adversarial examples of backdoored classifiers towards the target class, we expect the adversarial examples to resemble the backdoored instances. From our analysis in section 3.2 (Figure 2 and Figure 4), the adversarial examples indeed contain patterns that are similar to the original backdoor (in terms of color), which validates our expectation. Given the backdoor patterns observed in the adversarial examples, it is then natural to try to extract these backdoor patterns as new triggers and test their attack success rate, which is basically what we do in the paper.
>
> As for why the color patch and cropped patch we constructed have similar or higher success rates than the original trigger, we believe that this is an intrinsic property of backdoor poisoned classifiers. In other words, by introducing a secret backdoor trigger with a backdoor attack method, it also introduces potentially many alternative backdoors. What our approach does in this paper is to show the existence of these alternative backdoors. There could be other approaches to obtaining these alternative backdoors, but we believe our method is both conceptually simple and also effective.
>
> ** Relation to privacy **
> Our attack can be seen as uncovering a privacy issue in backdoor poisoned classifiers. We show how to extract alternative triggers despite the original backdoor being private. As for differentially private training algorithms, we believe it is addressing a different problem from ours. Differentially private training aims to make the final model leak as few information as possible on the training data. However, in our paper, we are not using the training data or trying to recover the training data. Naively applying differentially private training does not affect our attack as long as the trained classifiers are poisoned with a backdoor. We hypothesize that a variant version of differentially private training that specially aims to conceal the backdoor may be able to avoid our attack. We think this is an interesting open question.
>
> ** Robust Training **
> A recent work [2] investigated this setting. [2] show that adversarially robust models are still vulnerable to backdoor poisoning. Therefore, we believe that the phenomenon we observed in this paper for standard classifiers still exists for adversarial robust ones. In our updated version, we have added and discussed this work in the related work part.
>
> Thank you for your review again! If you have any further questions, we are happy to answer.
>
> [1] Robustness may be at odds with accuracy. Tsipras et al. ICLR 2019.
> [2] Exposing Backdoors in Robust Machine Learning Models. Soremekun et al. ArXiv 2020.

---

### Decision · Program_Chairs · 2021-01-07
**Final Decision**

**Decision:**

Reject

**Comment:**

The paper argues that a successful backdoor attack on classifiers is connected with further fundamental security issues. In particular they demonstrate and not only an original backdoor trigger but also other triggers can be inserted by anyone with access to the classifiers. Furthermore, the alternative triggers may appear very different from the original triggers, which confirms the claim in the paper's title that such classifiers are "fundamentally broken".

The paper offers an interesting insight into the features of poisoned classifiers. However, such insight is diminished by the fact that the proposed attack requires a substantial manual interaction. The user must manually analyze the adversarial examples generated for robustified classifiers in order to determine the key parameters of alternative triggers. While manual intervention as such does not undermine the main observation of the paper, this makes an automatic exploitation of this idea hardly feasible and hence decreases the significance of the paper's main result.